# Graphene-based synthetic antiferromagnets and ferrimagnets

P. Gargiani [1], R. Cuadrado[2,3], H.B. Vasili[1], M. Pruneda[2,3] & M. Valvidares [1]

Graphene-spaced magnetic systems with antiferromagnetic exchange-coupling offer exciting opportunities for emerging technologies. Unfortunately, the in-plane graphene-mediated exchange-coupling found so far is not appropriate for realistic exploitation, due to being weak, being of complex nature, or requiring low temperatures. Here we establish that ultra-thin Fe/graphene/Co films grown on Ir(111) exhibit robust perpendicular anti-ferromagnetic exchange-coupling, and gather a collection of magnetic properties well-suited for applications. Remarkably, the observed exchange coupling is thermally stable above room temperature, strong but field controllable, and occurs in perpendicular orientation with opposite remanent layer magnetizations. Atomistic first-principles simulations provide further ground for the feasibility of graphene-spaced antiferromagnetic coupled structures, confirming graphene's direct role in sustaining antiferromagnetic superexchange-coupling between the magnetic films. These results provide a path for the realization of graphene-based perpendicular synthetic antiferromagnetic systems, which seem exciting for fundamental nanoscience or potential use in spintronic devices.

[1] ALBA Synchrotron Light Source, Cerdanyola del Valles, E-08290 Barcelona, Spain. [2] Catalan Institute of Nanoscience and Nanotechnology (ICN2), CSIC and The Barcelona Institute of Science and Technology, Campus UAB, Bellaterra, 08193 Barcelona, Spain. [3] Universitat Autonoma de Barcelona, Cerdanyola del Valles, Bellaterra, 08193 Barcelona, Spain. Correspondence and requests for materials should be addressed to P.G. (email: pgargiani@cells.es) or to M.V. (email: mvalvidares@cells.es)

Antiferromagnetic (AF) systems, among which synthetic ferrimagnetic and antiferromagnetic (SFiM/SAF) structures, are receiving renewed attention for spintronic applications and magnetic information storage[1–3]. SAF structures based on metallic multilayers were initially developed in the 90s, inspired by the discovery of exchange coupling in multilayers and oscillatory magnetic interactions[4, 5]. A prototypical SAF structure is composed by two ferromagnetic films that are anti-ferromagnetically exchange-coupled through a non-magnetic spacing material due to the Ruderman–Kittel–Kasuya–Yosida (RKKY) interaction, and have been broadly used to improve the thermal and magnetic properties of spin valves[6–8]. Furthermore, exchange-coupled magnetic layers with perpendicular magnetic anisotropy (PMA) are intensively researched for the realization of vertical magnetic tunnel junctions (MTJs)[9], in views of potential application in future high-density and low current-induced magnetization-switching spintronic devices[10].

With the emergence of graphene and other two-dimensional (2D) atomic-crystals with unique electronic and structural properties[11], hybrid 2D-material magnetic systems are presently intensively investigated to enable new fundamental and applied

developments. Remarkably, graphene/magnetic structures display among other characteristics long spin-lifetimes at room-temperature[12], spin filtering[13, 14], and tunnel magneto-resistance[15–17], which are appealing properties for a range of innovative graphene-based spintronic technologies[18, 19]. Moreover, graphene has been reported to promote large PMA at the interface with magnetic thin-films[20, 21], thus possibly serving as a building-block for perpendicular spintronic devices incorporating a spacing layer with weak spin–orbit coupling. In this context, assessing the possibility to realize exchange-coupled PMA magnetic thin-films across a single-layer graphene (Gr) is of primary importance towards the realization of graphene spintronic devices. This has been recently stressed by a study of Yang et al.[21] with a theoretical prediction of strong PMA and AF exchange-coupling (AFC) in $Gr\{Co_n/Gr\}_m$ multilayers. However, the graphene-mediated exchange coupling playground remains largely unexplored, and graphene-spaced systems with large PMA and robust AFC have not been yet experimentally realized, in spite of their interest in future spintronic information processing technologies. In-plane AFC through a graphene layer was early observed between a Ni thin-film and Co-porphyrin molecules at

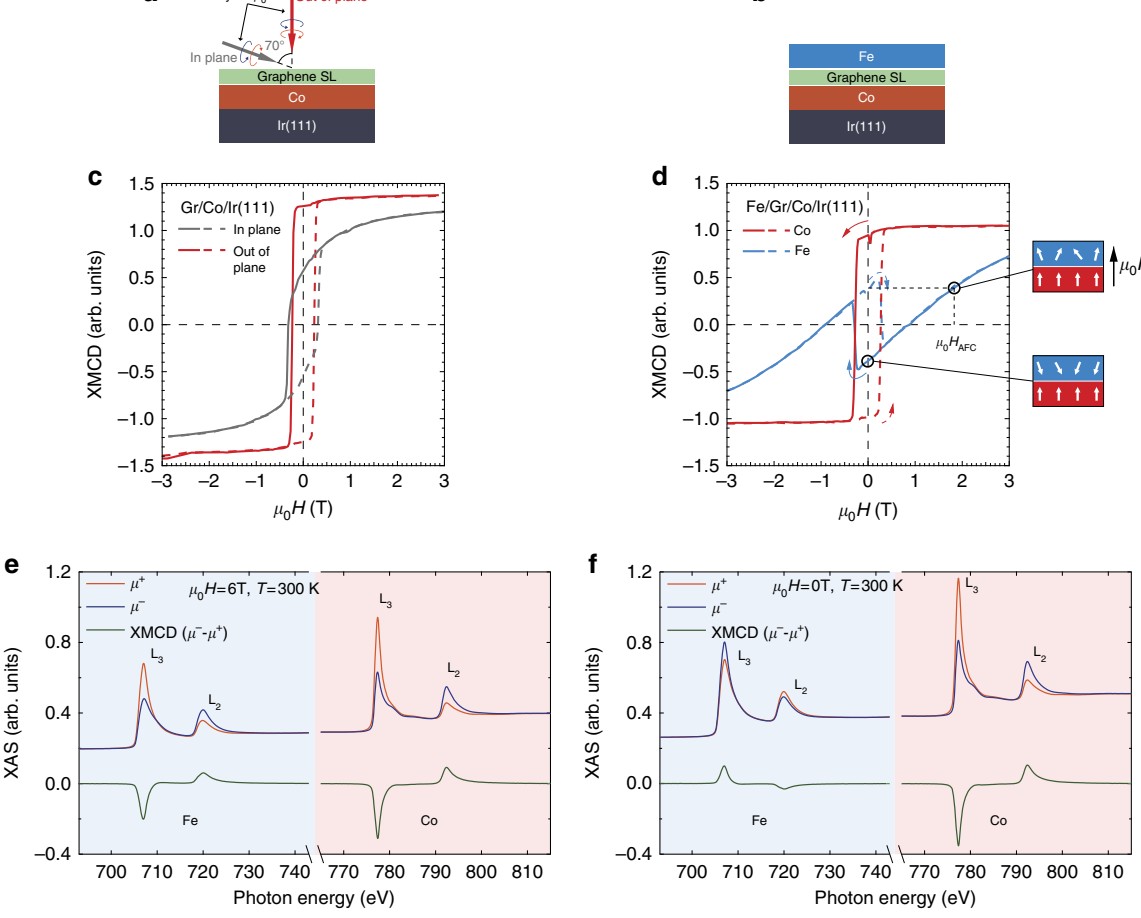

**Fig. 1** System properties as deduced by element-specific hysteresis loops and XMCD. **a** Experimental geometry for the in-plane and out-of-plane hysteresis measurements on the Gr/Co/Ir(111) and **b** a cartoon depicting the Fe/Gr/Co/Ir(111) multilayer sample. Element-specific hysteresis loops measured along the out-of-plane easy magnetization axis for a **c** Gr/Co[1.9 ML]/Ir(111) and **d** the same sample covered with Fe[1.6 ML]. Fe and Co loops were measured at $T = 300$ K as the field-dependent $L_3$ XMCD intensity, normalized by the pre-edge intensity value. An Fe magnetization of equal magnitude than its remanence value is enforced by an applied field $\mu_0 H_{AFC}$ onto a parallel configuration with the Co magnetization, as indicated in **d**, thus giving an estimation of the effective exchange coupling between the Fe and Co layers. **e**, **f** XMCD spectra measured on the Fe[1.6 ML]/Gr/Co[1.9 ML]/Ir(111) sample at $\mu_0 H = 6$ T and $\mu_0 H = 0$ T, respectively, after having previously magnetized the sample in an out-of-plane field of $\mu_0 H = +6$ T. The antiferromagnetic-ordering (i.e., antiparallel) of the Fe and Co layers magnetizations at remanence is directly evidenced by the sign change of the Fe XMCD **f** and in the element-specific hysteresis loop **d** between maximum and zero applied field

temperatures below ~200 K by Hermanns et al.[22], as well as in small-area MTJ devices with a graphene tunnel barrier and ferromagnetic metal electrodes by Li et al.[16]. More recently, graphene-mediated AFC between magnetic bulk single crystals and magnetic adatoms was studied, and it was shown to weaken or evolve onto a complex coupling when adatoms assemble to form small clusters[23]. Under these conditions, up-scaling for realistic applications is challenging.

Moving away from bulk magnetic materials into thin film structures or nanostructures is key to achieve the development of graphene-based robust AFC systems. The implementation of magnetic ultra-thin films allows to achieve strong magnetic anisotropies and exchange coupling via interfacial engineering, and also enables the introduction of symmetry breaking Dzyaloshinskii-Moriya and Rashba interactions[24, 25]. In contrast, the presence of a bulk magnetic layer dilutes the effect of interfacial contributions and eventually adds the complexity of magnetic surface closure domains. One suitable path to shift away from bulk magnetic materials is the intercalation approach widely used in the 70s–80s to tune graphite's physical properties[26]. In the last few years, it has been demonstrated that a number of elements can be intercalated between a graphene layer and its supporting substrate[27, 28], forming 2D layers that are well localized, chemically, and mechanically protected[29]. This approach was exploited to grow ultra-thin magnetic layers below graphene, which displayed a remarkable PMA enhanced by the interaction between graphene and the underlying magnetic metal thin-film[29–31].

Here, we show the realization of hybrid graphene-ferromagnet thin-film structures exhibiting strong perpendicular AF exchange-coupling. The Fe/Gr/Co/Ir(111) ultra-thin films investigated display a robust perpendicular AF exchange-coupling between the ferromagnetic layers mediated by the single-layer graphene spacer. Remarkably, the perpendicular AF coupling is stable above room temperature and field controllable, and displays a tunable remanent net magnetization as a function of the Fe layer thickness. Our first principle theoretical simulations confirm graphene's direct role in establishing the AF coupling via a graphene-mediated super-exchange mechanism. These results demonstrate the realization of graphene-based synthetic AF systems, which appear of fundamental interest in nanoscience and that possess an ensemble of properties well-suited for potential applications in spintronic devices.

## Results

### XMCD experiments.
A trilayer composed by an intercalated Gr/Co/Ir(111) ultra-thin structure (Fig. 1a) coupled to a Fe overlayer (Fig. 1b) displays a remarkable AFC at room temperature, as revealed by the sign switching of the Fe element-sensitive X-ray magnetic circular dichroism (XMCD) signal with and without an applied magnetic field observed in Fig. 1e, f, respectively (see Methods for further details). As deduced from the room-temperature field-dependent XMCD intensities reported in Fig. 1c, the Gr/Co[1.9 monolayer]/Ir(111) intercalated film presents high PMA and a large coercive field of $\mu_0 H_c = 0.27$ T, together with a 92% magnetic remanence along the out-of-plane easy-axis. The monolayer (ML) Co coverage was determined on the basis of Auger electron spectroscopy measurements (Supplementary Note 5). After the subsequent deposition of the Fe overlayer, the resulting Fe[1.6 ML]/Gr/Co[1.9 ML]/Ir(111) trilayer exhibits a room-temperature field-controlled AFC clearly evidenced by the Fe and Co element-specific hysteresis loops reported in Fig. 1d: as the applied magnetic field is decreased from the positive value of $\mu_0 H = +3$ T to 0 T, the Co magnetization (*red continuous line* in Fig. 1d) exhibits a high remanent

state, whereas the Fe magnetization (*blue continuous line* in Fig. 1d) is progressively reduced and eventually crosses zero at $\mu_0 H \simeq 0.8$ T; at lower fields the Fe magnetization reverses its sign yielding an antiparallel alignment at $\mu_0 H = 0$, as a result of the AFC between the Fe and Co out-of-plane magnetizations. For a negative applied field attaining the Co layer coercive field, the Co layer magnetization switches, driving the Fe layer magnetization on a corresponding abrupt jump, and preserving the AF alignment. Under increased negative field the Fe magnetization decreases monotonically, and eventually crosses zero aligning progressively with both the external field and the Co magnetization. In the reversed loop branch towards positive field, one finds again a remanent AF configuration with antiparallel magnetization and then attains positive coercive field, at which the Co layer reverses and drives once more the reversal of the Fe layer to maintain an AF coupled antiparallel configuration. For a larger external field that we define as the effective AFC field, $\mu_0 H_{AFC}$, the orientation of the Fe magnetization reverses and reaches a magnitude equal to its remanent value but now in parallel orientation with respect to the Co layer, thus overcoming the AF coupling interaction. This allows estimating the effective exchange-coupling energy density from the $H_{AFC}$ determined via the Fe magnetization loop as $J_{ex} = M_{Fe} \cdot t_{Fe} \cdot \mu_0 H_{AFC}$, which might be considered as a conservative estimate because using a field smaller than the saturation field. Still, using approximated values of the Fe magnetization and layer thickness for $M_{Fe}$ and $t_{Fe}$, we estimate $J_{ex} = 0.74$ mJ m$^{-2}$ at $T = 300$ K for the Fe[1.6 ML]/Gr/Co[1.9 ML]/Ir(111) sample. This is an exchange coupling strength of the same order of magnitude than that reported for conventional RKKY SAF multilayers[5].

Even at the maximum available external field of 6 T, the magnetic saturation of the Fe layer is not completely reached, as evidenced by the non-horizontal slope of the Fe magnetization curve. One likely hypothesis would be that this behavior is related to considerable 3D-film morphology, giving place to a distribution of switching fields and/or weekly coupled magnetic regions or grains. Other possibilities should not be excluded, taking into account that the magnetization reversal of AF exchange-coupled systems has been demonstrated to yield complex phenomenologies[32].

It is worth describing here in detail the Fe/Gr/Co/Ir(111) multilayer fabrication and optimization, as this path seems applicable to obtain further related FM/Gr hybrid structures with tailored properties. The growth of the Co film below the graphene layer was realized by the thermally activated intercalation of Co adatoms[29, 30], which were e-beam deposited on top of a single high-quality layer of Gr CVD-grown in-situ on Ir(111) (Methods section and Supplementary Note 3). In order to achieve high PMA, high remanent magnetization (above 80%) and coercivity, the optimal Co layer thickness was tuned at about $1.9 \pm 0.2$ ML (see Supplementary Notes 5 and 6). The cobalt intercalation was activated at temperatures exceeding 500 K, and optimal PMA/ high magnetic remanence of the Co layer was obtained at about $T = 700$ K. The Co saturation magnetic moment deduced via the XMCD sum rules[33] is $\mu = 2.25 \pm 0.07 \mu_B$, in agreement with previous reports[31]. The effective magnetic anisotropy energy (MAE) $K_{eff}$ for the Gr/Co[1.9 ML]/Ir(111) system is estimated employing the saturation field along the hard axis $H_{hard}$ and assuming a simple uniaxial anisotropy term, according to $H_{hard} = 2K_{eff}/\mu_0 M_{sat}$[24]. This yields $K_{eff} \lesssim 9$ MJ m$^{-3}$ and a MAE per-unit-area of $K_{eff} \cdot t \lesssim 2.6$ mJ m$^{-2}$, values in good agreement with previous findigs for the interfacial anisotropy of Co/Ir(111) and Gr/Co in the Gr/Co/Ir(111) system[30]. An enhancement of the PMA of the cobalt-graphene interface has been early reported by Vo-Van et al.[20] and later confirmed for Co-intercalated graphene systems[21, 30]. The intercalation process has been previously

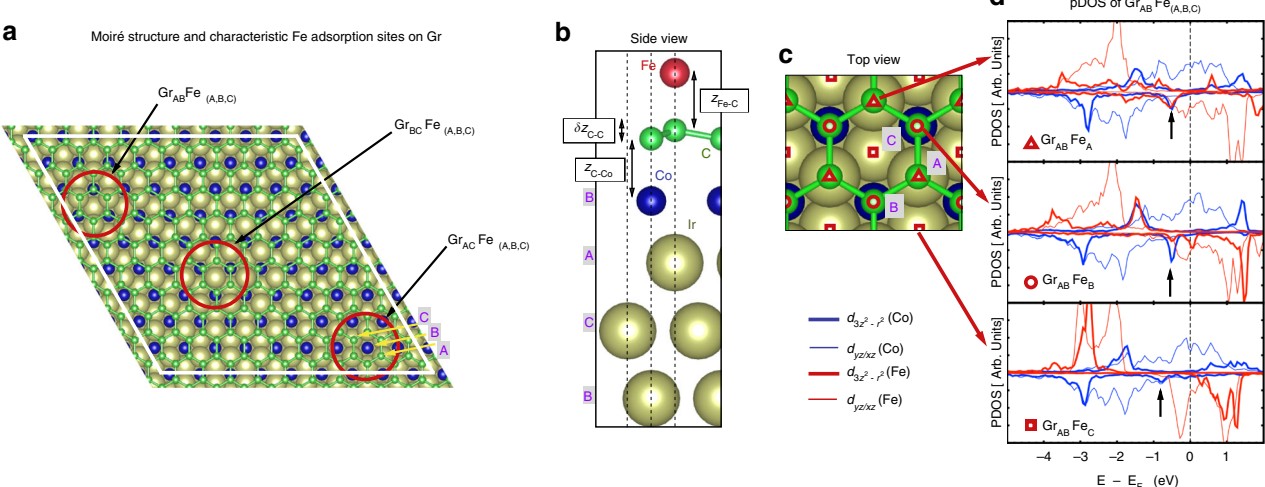

**Fig. 2** Structural models for Fe/Gr/Co/Ir(111) heterostructures. **a** Schematic top view of the Moiré superlattice defined by graphene on top of Co/Ir(111). Co is assumed to accommodate on the Ir lattice (in the simulations intermixing was neglected). Three distinct regions can be selected, that we name $Gr_{AB}$, $Gr_{AC}$, and $Gr_{BC}$, whose corresponding characteristic interlayer distances, adsorption energies, and exchange couplings are given in the Supplementary Table 1. **b** Side view of the $Gr_{AB}Fe_A$ configuration, and the stacking adopted throughout the paper for the Ir(111) lattice and for the different MLs' sites (A, B, and C). Co is always placed at the B site, with terminal Ir on the A site. For the selected commensurate lattice, there are two carbon atoms per Ir/Co/Fe, and we use the notation $Gr_{AB}$ to illustrate that the carbon atoms are placed on A and B sites. **c** Fe monolayer adsorption is defined either in A, B, or C sites, as shown in top view with *red triangles*, *circles*, and *squares*, respectively. **d** Projected density of states (pDOS) on Co and Fe $d_{3z^2-r^2}$ and $d_{yz/xz}$ for the three possible Fe stacks on top of $Gr_{AB}$

shown to take place preferentially at graphene point defects and wrinkles[29, 34], and in correspondence with Ir(111) step edges[31], remarkably preserving the structural integrity of the graphene layer. Our characterization by high-resolution scanning electron microscopy (SEM) (see Supplementary Note 4) and low energy electron-diffraction (LEED) (Supplementary Fig. 4) seems in good agreement with previously published reports, indicating the high quality of the Gr layer before-and-after the intercalation process. Moreover, by checking the protection against oxidation of the intercalated Co layer (Supplementary Note 9 and Supplementary Fig. 11), we obtained a further indication of the completion of the Co-layer intercalation and of the structural integrity of the graphene layer after the process. At a structural level, the sharp and low background LEED patterns (Supplementary Fig. 4) are indicative of a lattice-matched Co growth on Ir(111).

The LEED patterns provide also information on the growth morphology of the Fe overlayer in the Fe/Gr/Co/Ir(111) systems. This approach does not offer the capability of local probes with atomic resolution, but a unique statistically averaged information over a large area[35]. The Fe/Gr/Co/Ir(111) LEED patterns analysis (Supplementary Note 3) provides evidence for Fe islands having average lateral dimensions of the order of 6–8 nm for a 1 ML Fe film, suggesting that a large surface coverage can be attained already at few MLs coverage. These results seem to agree with what might be expected from our calculated adsorption energies, and are also in agreement with experimental findings obtained by STM on similar surfaces[36, 37]. The growth of Fe on graphene beyond an initial 3D growth phase, results on high-island density and high surface coverage (55% at 2 ML, which extrapolates to 80–90% for 3MLs) due to long-range repulsive interactions[37]. Our computed adsorption energies for Fe on Gr/Co/Ir(111) (see below) being comparable to the Fe on Fe ones, should equally favor the low-coverage nucleation of Fe islands over the incorporation of deposited atoms atop of the already-nucleated islands, favoring a large surface coverage. Summarizing the results of our LEED data analysis together with the adsorption-energy

interpretations, it seems reasonable to expect that the Fe films with equivalent thickness ranging between 2 to 4 MLs have a 3D thin film growth with a full or almost-full surface coverage, coexisting with 3D multilevel islands. It is interesting to mention that, as a result of the balance of adatom-adatom and adatom-substrate interactions, the growth morphology of Fe (and Co) layers on graphene appears controllable by deposition temperature and deposited thickness. Indeed, atomically-flat growth of magnetic overlayers on graphene has been demonstrated by pulsed laser assisted growth[20], and can be ascribed to a larger instantaneous growth rate favoring islands nucleation and reducing cluster formation. These considerations suggest that the morphology of such kind of AFC Gr-based multilayers might be tailored according to the requirements set by the desired application, which can be relevant for properties such as the coercive field or the exchange coupling[24, 38].

In relation to coercive field, magnetic anisotropy and magnetic remanence engineering, it is well-known that Co/Pd(111), Co/Pt (111), and other related systems such as Co/Ir(111) can display strong PMA with high coercivity and remanence[24, 39]. These properties can be controlled to a considerable extent by the thermal annealing temperature of the intercalation process and the total Co layer thickness. Thermally activated magnetic hardening in Co/Ir(111) films has been interpreted as a result of the partial interfacial alloying, with maximum coercive fields observed for annealing temperatures around 700 K[40], in close proximity to the intercalation temperatures used here. It is thus expected that a certain amount of Co atoms have diffused and intermixed at the very Co/Ir interface, as recently evidenced by a related surface X-ray diffraction investigation of the Co inter-calation on partially Gr-covered Ir(111)[41]. On the other hand, the preserved integrity of the graphene layer after Co intercalation (Supplementary Notes 3 and 7), avoids Fe/Co intermixing at the top Co or bottom Fe interfaces in room temperature deposition. Indeed, for a Fe-layer coverage of few ML's (up to 3–4 ML's) the Co magnetization loop is not significantly affected by the Fe deposition, showing the same XMCD signal at saturation and

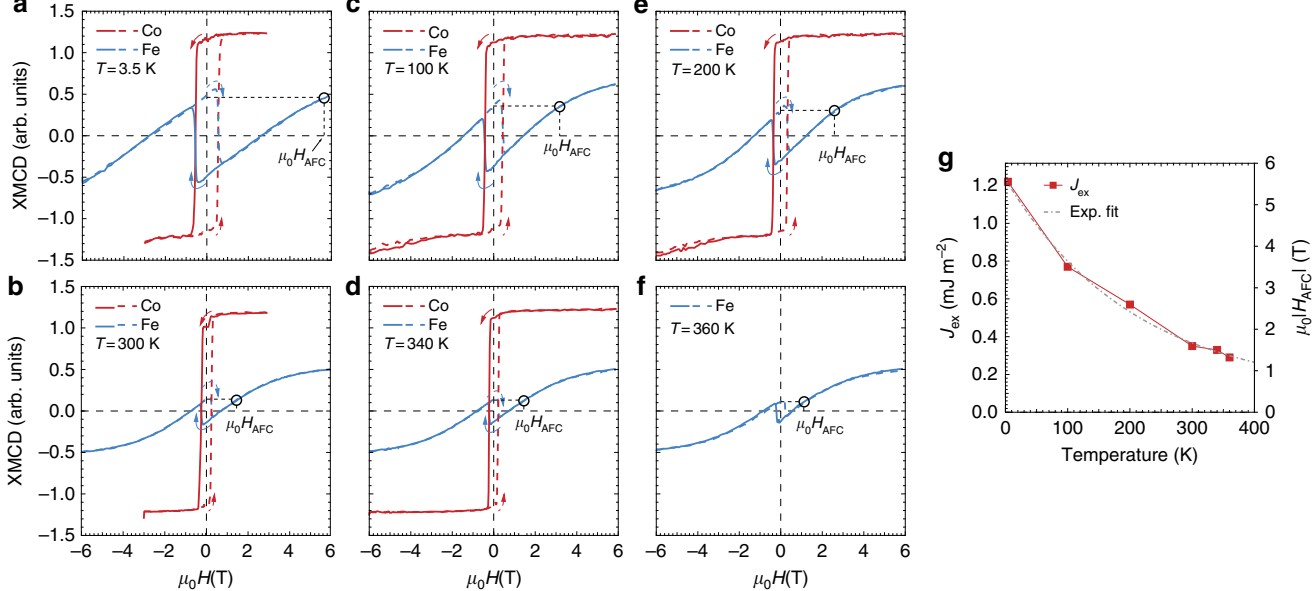

**Fig. 3** Temperature dependence of the antiferromagnetic coupling for a representative Fe[0.9 ML]/Gr/Co[1.9 ML]/Ir(111) sample. **a–f** Element-resolved (Fe, Co) easy-axis hysteresis loops collected at increasing sample temperature $T = 3.5, 100, 200, 300, 340,$ and 360 K respectively as the XMCD intensity maximum measured at the Fe and Co $L_3$ edge as a function of the applied field. We note that the Co loop at $T = 360$ K was not acquired, however the sign inversion of Fe loop still denotes an antiferromagnetic coupling between the Co and Fe layers. **g** Temperature-dependence of the experimentally-determined effective exchange-coupling energy density $J_{ex}$. Values are derived from the external field $\mu_0 H_{AFC}$ necessary to invert the Fe magnetization to a value comparable to the remanence one, according to $J_{ex} = M_{Fe} \cdot t_{Fe} \cdot \mu_0 H_{AFC}$, where $M_{Fe}$ is the bulk Fe magnetization and $t_{Fe}$ the Fe layer thickness

maintaining the same coercivity value as the bare Gr/Co/Ir(111). This evidences that the Gr spacing layer effectively preserves chemical and magnetic state of the layers, which is advantageous for engineering magnetic properties in a superposition scheme[14, 29].

**Atomistic calculations**. To investigate the nature of the coupling in the graphene-based AFC structure, we performed atomistic calculations based on density functional theory, and analyzed the role played by the single-layer carbon spacer. Our calculations are performed on a model for a Fe/Gr/Co ultra-thin film stacked on a Ir(111) surface, and reproduce both the AFC and the strong PMA observed in experiments. Due to the lattice mismatch between graphene and Ir(111), a Moiré pattern is expected in the heterostructure (the cobalt layer can be considered pseudomorphic with the Ir lattice), with a lattice parameter of ~2.5 nm (Fig. 2a)[42]. Rather than simulating this superstructure (with the $10 \times 10$ graphene unit cell and $9 \times 9$ Co/Ir lattice, which remains a challenge for atomistic first-principles calculations), we consider a computationally more efficient approach, where commensurability is assumed, and different stacking configurations are used to model the three principal Moiré domains sketched in the figure. On top of these, a Fe ML is considered, with Fe atoms placed on the possible A/B/C sites of the underlying Ir(111) lattice Fig. 2c. We obtain interlayer distances in good agreement with previous calculations for Co-intercalated in graphene[31, 43], as reported in Supplementary Table 1. We observe a strong corrugation of up to 1 Å in the Moiré lattice due to different $z_{C-Co}$ interlayer distances Fig. 2b. These lead to spatial variations of the couplings between magnetic layers, and different PMA energies. The theoretically calculated PMA energies for Fe/Gr/Co/Ir(111) interfaces, in the range of 2.2–4.1 mJ m$^{-2}$, seem in excellent agreement with our experimental observations.

The origin of the large PMA in Fe/Gr/Co/Ir(111) heterostructure is analyzed from the study of the magnetic anisotropy energy (MAE) at the different interfaces. The Co/Ir(111) and Gr/

Co/Ir(111) interfaces were studied by Shick et al.[43], who reported a strong effect of graphene on the anisotropy of ML Co on Ir (111), with an overall reduction of the MAE and a tendency for in-plane magnetization when carbon sits on top of Co. On the contrary, our calculations show that perpendicular magnetization is preferred for both Co/Ir(111) and Gr/Co/Ir(111) heterostructures, and although the strength shows some dependence with the stacking of carbon atoms (1.8, 1.3, and 0.0 mJ m$^{-2}$ for Gr$_{AB}$, Gr$_{AC}$, and Gr$_{BC}$, respectively), this affects marginally the already high MAE of ML Co on Ir (1.7 mJ m$^{-2}$). The MAE values resulting from our simulations are not far from the experimental values by Rougemaille et al.[30] (0.8 and 1.6 mJ m$^{-2}$ for Co/Ir and Gr/Co interfaces, respectively), or Vo-Van et al.[20] (an experimental estimation of 0.185 mJ m$^{-2}$ for Gr/Co interfacial MAE, and theoretical estimations of 1.0 and 1.2 mJ m$^{-2}$ for Co/Gr and Co/vacuum interfaces, respectively). A graphene-induced enhancement of the PMA has been reported for graphene-covered Co trilayers by Yang et al.[21], although their study do not include a Co/Ir(111) interface, nor different C stacks due to the underlying Moiré pattern. Our simulations for the case of ML Co on Ir(111) do not show a clear enhancement of the PMA due to graphene, but indicate a systematic increase in the computed PMAs when the Fe ML is placed on top of graphene, no matter what stacking (A/B/C) is considered.

In the following, we focus our attention on the Gr$_{AB}$/Co$_B$/Ir stacking, that has a short interlayer distance and a significant charge accumulation on graphene, suggesting that clustering of Fe adatoms in this region would be preferred during film growth[44] (similar results are obtained for the Gr$_{BC}$/Co$_B$/Ir stack). The computed Fe-ML adsorption energies, $E_{ads}$, evaluated after subtracting from the total energy of each configuration the energy of the clean Gr/Co/Ir slab and the Fe ML, conform to this assumption, with Gr$_{AC}$ being on average less favorable than Gr$_{AB}$ or Gr$_{BC}$.

For the three possible Fe-stacks considered, we obtain a strong interlayer exchange coupling, $J$, defined as the difference between

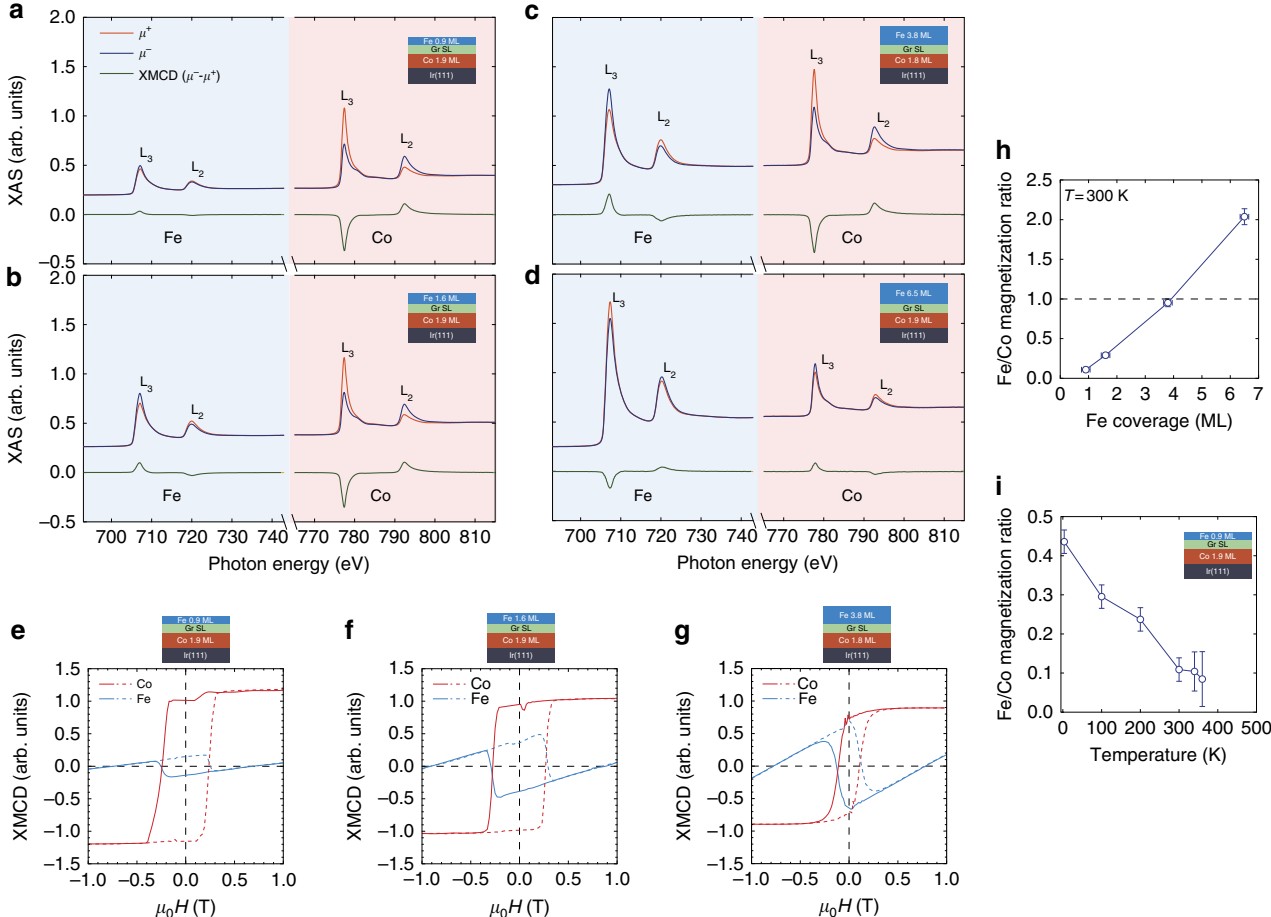

**Fig. 4** XMCD spectra and element-specific hysteresis loops for different Fe coverages at room temperature. **a–d** Fe and Co XMCD spectra collected in remanent state at increasing Fe-layer thickness, Fe[0.9 ML]-Co[1.9 ML], Fe[1.6 ML]-Co[1.9 ML], Fe[3.8 ML]-Co[2.1 ML], and Fe[6.5 ML]-Co[1.9 ML] respectively. The XMCD spectra are collected after applying a field of $\mu_0 H = +6$ T and subsequently ramping down to $\mu_0 H = 0$ T in order to fully magnetize the Co layer. The Fe XMCD-sign inversion in **d** indicates that the Fe moment direction is fixed by the external field rather than by the exchange coupling to the Co layer, nevertheless the AFC is still present as evidenced by the opposite sign of the XMCD between Fe and Co. Element-specific (Fe, Co) hysteresis loops measured as the XMCD signal at the $L_3$ absorption edge normalized by the pre-edge absorption signal as a function of the applied field for the **e** Fe [0.9 ML]-Co[1.9 ML], **f** Fe[1.6 ML]-Co[1.9 ML, and **g** Fe[3.8 ML]-Co[2.1 ML] samples. The sample reported in **g** shows a slightly lower coercive field as a result of the lower thermal-induced Co intercalation temperature used, nominally $T = 640$ K as compared to $T = 700$ K. **h** Fe coverage-dependence of Fe/ Co magnetization ratio as estimated via the XMCD sum rules (data relative to spectra reported in **a–d**). **i** Temperature dependence of the Fe/Co magnetization ratio measured on the sample Fe[0.9 ML]/Gr/Co[2.1 ML]/Ir(111). The error bars on the magnetization ratios of **h**, **f** reflect the uncertainty in the background estimation for the XMCD sum rules analisys; the error bars on the Fe coverage **h** mirror the systematical error in the coverage determination

the energies of parallel (FM) and antiparallel (AF) alignments of magnetizations in Fe and Co ($J = E_{FM} - E_{AF}$). Resulting exchange coupling energies depend on the stacking and Fe adsorption site on top of graphene, with values between 15 and 277 mJ m$^{-2}$ as indicated in Supplementary Table 1, larger than the values predicted for symmetric Co/Gr/Co or Fe/Gr/Fe junctions[45]. Similarly strong couplings are obtained for Gr$_{BC}$/Co$_B$ stacks, while the coupling is substantially reduced (even switched to FM) for Gr$_{AC}$/Co$_B$, where the interlayer distance is larger, and the charge accumulation on graphene suppressed. We note that a fair comparison with the experimental estimate requires an average over the whole Moiré pattern, considering temperature effects, and taking into account possible impurities and irregularities at the interface, all of which would result in a significantly reduced value.

The mechanism of the AF coupling between Fe and Co can be tracked down from the analysis of the electronic structure of the heterointerface. Figure 2d shows the projected Density of States

(pDOS) on the metallic $d$ states that point out of the layer plane ($d_{3z^2-r^2}$ and $d_{yz/xz}$). It is known that hybridization with graphene's $2p_z$ states strongly affects the energy position of Co-3$d$ states[21, 46]. A $d_{3z^2-r^2}$ peak at ~0.5 eV below the Fermi level is apparent for both Co and Fe in all atomic arrangements that give larger couplings, and its amplitude correlates with the value of the $J$ couplings. Notably, if we take the same fixed structures but removing the graphene ML (Supplementary Note 1 and Supplementary Fig. 1), the peaks disappear, revealing that superexchange through C-$p_z$ states is key (Supplementary Fig. 2). This confirms a direct role of the graphene 2D-layer in sustaining AF superexchange-coupling between the two magnetic films, in line with earlier propositions by Hermanns et al.[22], Barla et al.[23], or Yang et al.[21] among others. Indeed, for the Co–Fe interlayer distances obtained, the coupling becomes FM in absence of graphene, and reduces by an order of magnitude. Furthermore, calculations done for bilayer and trilayer graphene spacers result in tiny but FM coupling, suggesting that the behavior of the

spacer departs from that of a semimetal, where AF couplings with long decay lengths are obtained[47]. The strong distortion of graphene's electronic structure due to the hybridization with the transition metals can be related to this observation. The remarkable buckling $\delta z_{C-C}$ in the graphene ML is another consequence of this hybridization.

**Stability of AF coupling and compensated configurations.** Having presented both experimental and theoretical evidences supporting the realization of graphene-mediated AFC in ferromagnet-Gr structures, we move on to discuss the robustness of this AFC behavior against temperature or layer-thicknesses variations. The set of Fe and Co magnetization hysteresis loops in Fig. 3a–f, covering a wide temperature range from $T = 3.5$ K up to $T = 360$ K, reveals a remarkable stability of the AFC state for a Fe[0.6ML]/Gr/Co[1.9ML]/Ir(111) sample grown at room temperature. As the temperature is lowered, an increase in the out-of-plane Fe layer remanent magnetization, together with a notable increase in the Co coercive field, is observed. The exchange coupling $J_{ex}$ and $\mu_0 H_{AFC}$ field (see Fig. 3g) both increase at lower temperatures, indicating that the AFC state becomes more robust. Nonetheless, the AFC state persists at temperatures up to 360 K, as demonstrated by the $J_{ex}$ and the $\mu_0 H_{AFC}$ field vs $T$ shown in Fig. 3g. These temperature trends might result from increments in the magnetic anisotropy, the magnetic susceptibility and magnetic moment of the Fe layer, the exchange coupling strength, or a combination of several of these factors.

To assess the feasibility of a magnetically compensated graphene-based SAF/SFiM system, we analyzed the room temperature dependence of the magnetic properties of a Fe/Gr/Co/Ir(111) heterostructure as a function of Fe top-layer thickness. We take special care in investigating two scenarios: the first, a magnetically compensated system at room temperature; and the second, a structure whose Fe top-layer magnetic moment exceeds that of the Co layer, providing an inverse scenario to that of an AF structure with a dominant PMA Co layer so far discussed. In Fig. 4a–d we report a selected set of Fe and Co $L_{2,3}$ XMCD spectra, which were collected at $T = 300$ K in remanent state for increasing Fe layer nominal coverage on Gr/Co[1.9–2.1 ML]/Ir (111) films. The AFC remains stable among the different coverages, as evidenced by the XMCD-sign inversion between Fe and Co atomic edges. Most notably, the X-ray absorption (XAS)-normalized Fe XMCD signal measured at $\mu_0 H = 0$ increases monotonically with Fe coverage up to a 3.8 ML Fe layer thickness. This indicates that the remanent Fe magnetization depends on the Fe layer thickness, as evidenced by the element-specific hysteresis loops shown in Fig. 4e, f. In order to assess the relative weight of the Fe and Co remanent-state out-of-plane magnetization at different layer thicknesses and temperatures, we applied the XMCD sum rules[33]. The XMCD sum rules can give quantitative information on the atomic magnetic moment of the probed atom. We can estimate the relative Fe and Co magnetization for the Fe/Gr/Co/Ir(111) system as $M_{Fe}/M_{Co} = t_{Fe}\mu_{Fe}/t_{Co}\mu_{Co}$, where $t_{Fe}$, $t_{Co}$ are the layer thicknesses, and $\mu_{Fe}$, $\mu_{Co}$ are the atomic moments of Fe and Co obtained via the XMCD sum rules. In Fig. 4h we report the room-temperature $M_{Fe}/M_{Co}$ ratio as a function of the Fe layer thickness. The data points are relative to the spectra reported in Fig. 4a–d. The $M_{Fe}/M_{Co}$ room-temperature ratio shows a clear dependence on the Fe film thickness for thicknesses below ~10 ML's and notably crosses the magnetic compensation condition ($M_{Fe}/M_{Co}$ = 1) for values close to Fe ~4 ML's. This result provides an indication that the total magnetization can be controlled by tuning the Fe layer thickness, demonstrating that a Fe/Gr/Co/Ir (111) magnetically compensated system can be realized at room

temperature. A strong variation of the $M_{Fe}/M_{Co}$ ratio is also observed as a function of sample temperature, as reported in Fig. 4i for the Fe[0.9 ML]/Gr/Co[1.9 ML]/Ir(111) sample. Altogether, these observations prove that it is possible to achieve a determined magnetization ratio in a chosen temperature range by tuning the sublayer's thicknesses.

For Fe thicknesses sufficiently larger than that of the Co layer, the $M_{Fe}/M_{Co}$ ratio takes values higher than unity, indicating that the out-of-plane remanent magnetization of Fe layer exceeds that of the Co layer. The Fe out-of-plane remanent magnetization (see Fig. 4d) conserves the direction of the previously applied maximum external field, as deduced by the sign of Fe XMCD. In contrast, the Co remanent magnetization is now the one switching and orienting antiparallel to the Fe magnetization, presenting an XMCD-sign reversal (see also Supplementary Figs. 7 and 9). This observation is in agreement with a preserved AFC between the Co and Fe layer but with a magnetostatic energy balance favoring the Fe top-layer, that drives the magnetization direction of the Co bottom-layer. In this situation we observe an appreciable reduction of the Fe and Co remanent magnetizations as deduced from the XMCD intensity in Fig. 4d, which might be related to the formation of magnetic domains in the Fe layer and that, via the AF exchange coupling, imprint on the Co layer.

## Discussion

The experimental and theoretical results presented above demonstrate that it is possible to design Gr-based SFiM structures with a vanishing net remanent magnetization at a determined compensation temperature, i.e., SAF systems. Such graphene-based SAF structures are appealing for having a closed-flux magnetization configuration with a very low, ultimately vanishing, and stray macroscopic magnetic field. This minimizes the demagnetization energy, defining a magnetic configuration of high stability. Hence, altogether, the studied Fe/Gr/Co/Ir SAF structures gather a notable number of appealing magnetic properties, among which we highlight: a perpendicular easy axis with strong magnetic anisotropy; a strong AFC along the perpendicular direction; stability of AFC against field and temperature; robustness of magnetic properties and AFC with layer thickness; tunability of structures from uncompensated ferrimagnetic to essentially compensated AF configurations.

When compared to commonly used in-plane SAF spin valve structures such as CoFe/Ru/CoFe[8] or CoFeB/MgO(Ru)/CoFeB MTJs[9, 48], graphene-based SAF systems present analogies but also differentiating characteristics and potential advantages by merging appealing magnetic properties with graphene unique electronic, mechanical, or thermal properties. First, the Gr spacing layer can contribute to the definition or preservation of the chemical/magnetic interfaces, acting as a barrier for diffusion preventing alloying of two neighboring materials or protecting a magnetic material[14], in analogy to MgO(1–2 nm) or $Al_2O_3$ barriers[6]. Second, a single Gr spacing layer has been shown to perform as an effective pinhole-free spacing layer for the fabrication of MTJs, in spite of its single atom-layer character[15, 16]. Also, the graphene-mediated AFC between two layers allows a SFiM or SAF structure avoiding the need for a high spin–orbit metallic barrier such as Ruthenium, or the need of a third layer for the biasing as in MTJs with MgO or $Al_2O_3$ barriers. And finally, graphene-based perpendicular SAF hybrid structures could exploit graphene unique properties such as its outstanding mechanical and thermal characteristics in the design of flexible electronic devices[49] or improving thermal management strategies in devices[50].

It is worth briefly discussing potential paths and challenges for the eventual technological introduction of these structures on

devices. The monolithic integration of graphene on silicon has been recently reported via an Yttrium-Stabilized-Zirconia (YSZ) buffer approach. Remarkably, the growth of Gr on Ir(111)/YSZ/Si (111) substrate is scalable in wafer size up to 4-inches and the Gr layer quality is comparable to that obtained for the CVD growth on Ir(111) single crystals[51]. Hence, a YSZ/Si(111) platform appears a relevant and feasible path for the scalability and monolithic silicon integration of the SAF structures investigated in this work. Regarding compatibility with device fabrication processes such as complementary metal-oxide-semiconductor (CMOS), the high CVD Gr growth temperature here used (i.e., 1300 K) exceed the thermal budget of CMOS manufacturing, being high enough to produce issues by diffusion or melting. Alternative graphene-growth methods such as plasma assisted[52], CVD reactors[14], or polycyclic aromatic hydrocarbons precursors[53, 54] enable a significant reduction of graphene-growth temperatures (700–900 K). Alternatively, in past years an increasing number of solutions have been proposed for hetero-geneous Gr-Si integration, employing the clean transfer of Gr grown on other supports directly onto $SiO_2$/Si or silicon-on-insulator substrates[55, 56]. This has enabled roll-to-roll production of 30-inch graphene films for transparent electrodes[57], graphene Si-CMOS hybrid Hall integrated circuits[58] or Gr/CMOS inte-grated image sensors[59], among others. It is also noteworthy that Gr-based flexible devices have been demonstrated by alternative fabrication routes to CMOS[49, 60]. The field is in continuous development as a result of the strong efforts being deployed for addressing the challenges towards graphene spintronic devices[19].

In summary, we have addressed the feasibility of Fe/Gr/Co/Ir (111) structures displaying a strong perpendicular AFC that is robust on temperature and Fe layer thickness. Atomistic calcu-lations confirm graphene's direct role in sustaining AF superexchange-coupling between the magnetic layers, and are in good correspondence to the experimental findings. These results demonstrate an additional class of synthetic-AF multilayered materials that, while being of fundamental interest, appear cap-able of providing practical magnetic devices with PMA which are potentially relevant for perpendicular magnetic recording media, perpendicular spin valves, MTJ structures[3, 9], or in all-optical switching magnetic materials[2, 61]. We expect that these results will help to spark interest towards the search and dis-covery of related perpendicular AFC Gr-based magnetic multi-layers, a class of materials largely unexplored and unexploited at present, but which could enable developments in the field of graphene spintronics.

## Methods

**Sample fabrication.** The samples were all prepared in situ in the preparation chamber available at the XMCD magnet endstation of the BOREAS beamline at the ALBA synchrotron, with a base pressure better than $1 \times 10^{-9}$ mbar. The Ir(111) single crystal was prepared by repeated cycles of $Ar^+$ sputtering at 2 KeV followed by annealing at $T = 1000$ K. The quality of the surface was checked by LEED, giving a sharp six-fold hexagonal pattern without any presence of reconstructions or diffuse background. The Gr/Ir(111) was then prepared exposing the clean substrate held at $T = 1300$ K to a $C_2H_4$ residual gas atmosphere at a pressure of $2.0 \times 10^{-6}$ mbar for 10 min. This procedure leads to the formation of large single-domain single-layer graphene over the whole surface area, as has been reported[42] and as deduced from the Moiré LEED pattern (Supplementary Fig. 4). The Gr/Ir(111) samples kept at room-temperature were exposed to a Co atomic flux evaporated from a high-purity rod by electron bombardment. The Co deposition rate was about 1 Å min$^{-1}$ as determined by a quartz micro-balance. The intercalation process was done either in a single step, after having deposited the full amount of Co, or in several intercalation steps at temperatures in the range 570–700 K. Although all the films were showing large PMA and remanent magnetization, larger coercivity values were observed for films prepared either in several steps or at higher annealing temperatures. The full intercalation of Co below the graphene layer was checked by exposing the Gr/Co/Ir(111) to molecular oxygen at a pressure of $1 \times 10^{-6}$ mbar for 5 min an then checking the XAS at the Co $L_{2,3}$ edge. In case of non-completed intercalation the sample was showing definite signs of Co oxidation (Supplementary Note 9), while for complete intercalation the sample was showing

a pristine Co $L_{2,3}$ absorption edge. The Fe was also deposited at room temperature by electron bombardment evaporation from a high-purity (99.999%) rod.

**XMCD measurements.** The XAS and magnetic circular dichroism experiments were carried out at the BOREAS beamline of the ALBA synchrotron using the fully circularly polarized X-ray beam produced by an apple-II type undulator[62]. All measurements were performed in situ immediately after sample preparation. The base pressure during measurements was ~$1 \times 10^{-10}$ mbar. The X-ray beam was focused to about $500 \times 500$ μm$^2$, and a gold mesh has been used for incident flux signal normalization. The XAS signal was measured with a Keythley 428 current amplifier via the sample-to-ground drain current (total electron yield TEY signal). The magnetic field was generated collinearly with the incoming X-ray direction by a superconducting vector-cryomagnet (Scientific Magnetics). The magnetization loops were measured sweeping continuously the magnetic field at a fixed speed and acquiring the absorption TEY current at the maximum of the $L_3$ XMCD signal and at a pre-edge position in order to cancel-out any field-induced artifact in the measurements. We recall that such XMCD measurements at the Co and Fe $L_{2,3}$ absorption edges provide direct element-specific information on the magnitude and sign of the projection of Co and Fe magnetizations along the beam and field direction.

**Atomistic calculations.** Our density functional based calculations were performed using the SIESTA code[63]. The generalized gradient approximation[64] for the exchange-correlation (XC) potential was considered. We used norm-conserving pseudopotentials in the separate Kleinman-Bylander[65] form under the Troullier-Martins parametrization[66], and to address a better description of the magnetic behavior, nonlinear core corrections were included in the XC terms[67]. The geometry optimizations were carried out using the conjugate gradient method at spin-polarized scalar relativistic level. A double-$\zeta$ polarized with strictly localized numerical atomic orbitals was used as basis set, and the electronic temperature-$k_BT$ in the Fermi-Dirac distribution-was set to 5 meV. The relaxed lattice parameter for the Ir(111) surface was 2.715 Å, in very good agreement with previous experimental reports. After the relaxation process the forces per atom were less than 0.01 eV Å$^{-1}$. Based on the locality of the fully relativistic contribu-tion, there are two different levels of approximations when the spin–orbit (SO) is taken into account. We have used in the present work the off-site approach[68], that takes into account not only the local SO contributions to the total energy but also the neighboring interactions between atoms to obtain the total self-consistent energy. As usual, the MAE is defined as the difference in the total self-consistent energy between hard and easy magnetization directions. Within the present work, we performed an exhaustive analysis of the MAE convergence in order to achieve a tolerance below $10^{-5}$ eV. We employed around 1000 k-points in the calculations for each geometric configuration, which was sufficient to achieve the stated accuracy (Supplementary Note 2 and Supplementary Fig. 3).

**Data availability.** The data sets generated during and/or analyzed during the current study are available from the corresponding authors on reasonable request.

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

## Acknowledgements

The research leading to this work has been funded by Spanish MINECO/FEDER, grants no. FIS2013-45469-C4-3-R (AEI/FEDER, UE), FIS2015-64886-C5-3-P, FIS2016-78591-C3-2-R (AEI/FEDER, UE), and MAT2014-59315-R, Generalitat de Catalunya (2014SGR301), CERCA Programme/Generalitat de Catalunya, and by European Union H2020-EINFRA-2015-1 program under grant agreement No. 676598 project "MaX—materials at the exascale". ICN2 is supported by the Severo Ochoa program from Spanish MINECO (Grant No. SEV-2013-0295). P.G., H.B.V. and M.V. acknowledge additional support via ALBA IHR program. M.V. and P.G. also acknowledge Dr. A. Scholl (LBNL) for helpful discussions and M. Rosado (ICN2) for technical assistance with high-resolution SEM. R.C. acknowledges the funding from the European Union's Horizon 2020 research and innovation program under the Marie Skłodoswa-Curie grant agreement no. 665919.

## Author contributions

M.V. and P.G. conceived the experiments. P.G., H.B.V. and M.V. performed sample growth and LEED, Auger, XAS, and XMCD experiments at ALBA BL29. R.C. and M.P. designed the theoretical models and performed atomistic calculations. P.G. and M.V.

analyzed and interpreted XMCD data. P.G., M.V., R.C. and M.P. discussed the magnetic behavior. P.G., M.V., M.P. and R.C. prepared the manuscript. All authors read and commented on the manuscript.

## Additional information

**Competing interests:** The authors declare no competing financial interests.

