## [Peer Review File · Nature Communications]

Reviewers' Comments:

Reviewer #1 (Remarks to the Author):

A file is attached.

Reviewer #2 (Remarks to the Author):

This paper reports the existence of a robust antiferromagnetic exchange interaction at room temperature across a single layer of graphene in perpendicularly magnetized Ir/Co/graphene/Fe multilayers. This is based on measurements of element-selective magnetic hysteresis using XMCD, which are quite convincing. The paper also models the system using density functional theory and claims results in good qualitative agreement with theory. This is fine original work, and an additional materials family capable of providing practical magnetic devices with perpendicular magnetic anisotropy would certainly be welcome. However, in my opinion, the paper does not yet make a persuasive case for why these results are sufficiently important to justify publication in Nature Communications. In both the abstract and concluding paragraph, the paper argues that the results represent “unprecedented magnetic properties” that appear “suitable for application”. However, the justifications given in the paper for these two claims are weak. I suggest that the authors consider the following questions and comments.

1. Regarding “unprecedented magnetic properties”: Many different spacer layers can provide the antiferromagnetic coupling at room temperature needed for making synthetic antiferromagnetic devices (e.g., Parkin, Phys. Rev. Lett. 67, 3598 (1991)). In what way, precisely, are the magnetic properties of the graphene devices unique? For example, if the thicknesses of the magnetic layers are scaled to make an honest comparison, how does the strength of the antiferromagnetic exchange coupling in the graphene devices (i.e., in erg/cm²) compare to what has been measured previously? The paper under review never considers this central point. If the strength falls in some new regime, what is the practical significance? The paper also tries to make “ultimately-thin” devices a selling point, but existing magnetic multilayer structures are already routinely made with layer thicknesses of 1 nm and below. Why is going down to single atomic layers advantageous or important?

2. Regarding “appearing suitable for application”: This point seems dubious to me, as well. To be suitable for applications, a magnetic multilayer needs not only to have excellent magnetic properties, but it must also be capable of integration into devices with large magnetoresistance to allow read-out of the magnetic state. The only practical way of doing this at present is to make MgO-based magnetic tunnel junctions, which generally requires the use of sputtered FeCoB

films with an annealing step. The use of pure Fe or Co, and the rough 3-D island growth mechanism of the Fe layer both suggest that integration into practical high-magnetoresistance tunnel-junction structures will be difficult or impossible.

3. In the concluding paragraph, the authors make a claim about being “silicon integrable”. This is not a fair claim, because the fabrication procedure involves graphene growth at 1300 K. This is far beyond what is possible for back-end processing of Si-based devices. Even the temperature of the intercalation step (570 – 700 K) could be problematic.

4. The authors express the results of the DFT calculations for the magnetic anisotropy and the antiferromagnetic exchange coupling in terms of energies (2.1 – 4.5 meV for the PMA and 255 meV for the AFC). However, these quantities are really energies per unit area. What is the relevant area for interpreting these calculations?

5. The authors express the results of the experimental measurements for the magnetic anisotropy and the antiferromagnetic exchange coupling in terms of effective magnetic fields (in Tesla). They should provide the translation between these units and the energy per unit area determined in the DFT calculations so that readers can make comparisons between the two types of results.

6. A strong antiferromagnetic exchange interaction across single-layer graphene has been observed previously for relatively-thick in-plane-magnetized magnetic layers (Li et al., Phys. Rev. B, 89, 184418 (2014)). The paper under review is different in dealing with much thinner magnetic layers with perpendicular magnetic anisotropy, but they should still consider referencing this previous work.

7. The paper contains several awkward uses of English. These are relatively minor in that they do not affect understanding the work, but the paper could benefit from further editing in this regard. The level of detail provided about the experimental protocols is good.

Reviewer #3 (Remarks to the Author):

The authors presented both experimental and theoretical results showing antiferromagnetic coupling (AFC) along with perpendicular magnetic anisotropy (PMA) in Fe/Gr/Co on Ir(111) structures which can serve as synthetic antiferromagnet (SAF) for spintronics. While the findings are interesting and represent a certain degree of novelty, it is hard to find them really breakthroughing in a view of previous reports in the literature including those the authors cite in the manuscript. In fact, some of the references are cited "superficially" even though they should be referred more appropriately. I think a major revision of the text, especially the introduction

and conclusion sections, is needed prior to be accepted for publication in order to show/highlight the added breakthrough in the field by citing in appropriate context previous works in the topic and clearly reveal the originality of the results presented. The following points should be addressed and/or clarified during the revision:

1) The concept of SAF, or sometimes called artificial antiferromagnets (AAF) is well-known and widely used in spintronics since 1990s when they were proposed as a reference layer for instance in spin-valve structures and not simply in the context of RKKY interlayer exchange AFC led to the GMR etc. authors refer which creates a confusion with GMR layered structures which are not at all SAF. This confusion should be eliminated and SAF importance in their appropriate context along with appropriate references should be given in introduction.

2) The importance of PMA for spintronics is missing and should be highlighted as well. Also, when discussing the results in Fig. 1 (lines 43-49), the authors should clearly cite here Ref. 28 (C. Vo-Van et al, NJP 2010) as one of the first with PMA report in Gr/Co interfaces.

The authors stated that they followed the intercalation approach to fabricate their structures so I suggest to cite also the work by J. Coraux et al [J. Phys. Chem. Lett. 3, 2059 (2012)] which is important in this context and comment/compare.

A minor comment is Fig. 1(a),(b) and (e) should be referred in the text.

3) When presenting theoretical results in lines 137-138, the authors should clearly state that the AFC and PMA calculations results for Fe/Gr/Co reproduces not only their experiment, but these results reproduces and confirms earlier proposition of the same concept of Co/Gr/Co heterostructure with strong AFC and PMA by Yang et al [Nano Lett. 16, 145 (2016)] in order to boost the effective PMA. Even though authors cite this work later in the text in the different context (Ref. 36), the primary result of that work is very similar to the one proposed here and was also featured in

<http://phys.org/news/2015-12-giant-magnetic-effect-benefit-spintr onics.html>

The authors should compare their findings with this work.

4) When presenting theoretical results in Fig. 2 for PMA in Gr/Co/Ir, the authors should precise if 2.1-4.5 meV range of anisotropy values are per atom or per cell and clearly confront also these values with those in earlier reports including Refs. 36 and 28 in addition to Ref. 33. In particular it would be interesting to compare the most stable stackings in different works. Also, it is very interesting to discuss whether the PMA in their structure is provided by Gr/Co or Co/Ir interface and this should be discussed also in comparison of earlier works.

5) I found the discussion of XMCD results in Figs. 3 and 4 a little too vague including the link between temperature dependence and Fe and Co magnetization relative role, it should be

rewritten more clearly.

6) The authors performed the MAE calculations with 1000-k points using SIESTA package. Since it is a recent implementation in this code, a clear demonstration of convergence as a function of k-points number is needed.

All these points should be addressed in the revised version of the manuscript in order to improve the quality of the manuscript and clearly underline the originality of the results presented not by ignoring previous works but by comparing with them. It is difficult to agree without this revision that the authors demonstrated unprecedented magnetic properties as they claim in the conclusion with a superexchange coupling which they claim original even though it was already proposed theoretically for Co/Gr/Co structures in ref. 36.

So the conclusion should also be revised.

There are also many typos/corrections needed:

Line 41: TMJ?

Line 209: PMI?

Fig. 3: a), b), c), d), ... are missing and temp. dependence should be h)?

Lines 251-252: SFI?

Comments Reviewer #1:

This study reports on experimental realization of a strong perpendicular antiferromagnetic exchange coupling (AFC) between two thin Fe and Co layers separated by a monolayer graphene, along with a numerical analysis on the AFC mechanism. The Co film was grown below the graphene layer by the thermally activated intercalation of e-beam deposited Co adatoms between a CVD-grown single layer of graphene and a Ir(111) substrate. The established AFC, occurring perpendicular to the multilayer plane with the high remanent layer magnetization, is quite insensitive to the thickness variation of the Fe overlay and robust to the temperature increase up to room temperature. The tunability of the AFC by an external B field can be conveniently utilized for device applications of magnetic information storage and/or spintronics. First-principle simulations reveal that the superexchange coupling between Fe and Co atoms through C- p_z states, which is provided only in the presence of monolayer graphene, is essential to the AFC.

The experiments were nicely performed including the essential points of examination together with the relevant numerical analysis. Although closely related studies on synthetic ferrimagnet and synthetic antiferromagnet structures already exist [JAP 99, 08B703 (2006)], this study constitutes the first graphene-based study on the similar magnetic functionality. It also provides details of the device fabrication, stability of the magnetic properties and its control, and the theoretical analysis as well, which will be much beneficial to the ones working in the related fields. Thus I can recommend accepting the manuscript for publication in Nature Communications. Nonetheless I believe the following points should be complemented before its full acceptance.

- Although authors provide surface analysis results to persuade of the high quality of their Co and Fe thin films, there is no picture showing directly their surface morphology. The STM topography, AFM or at least SEM microscopy should be provided maybe in the Supplementary Information. It is mentioned in the main text that the Co adatom intercalation took place at graphene point defects and wrinkles. This seems to suggest that much disorder was present in the graphene layer, which would have certainly been even more degraded after the Co intercalation. Thus, to validate the numerical calculation for the ideal situation more information should be provided for the surface state of the monolayer graphene right after the Co intercalation. Information should include the monolayer coverage of the graphene layer and the distribution of its different domains, which, according to the numerical calculation, is essential to generating the desired antiferromagnetic superexchange coupling between Co and Fe layers. What was the range of examination for the LEED patterns shown in Fig. S3 in comparison with the average domain size of the graphene layer?

- As mentioned above, a previous publication [JAP 99, 08B703 (2006)] reports on closely related studies on synthetic ferrimagnet and synthetic antiferromagnet structures, although graphene was not adopted for the study. This work should be cited with brief mentioning on the difference between the two studies in comparison.

- The description in the manuscript in general is very loose and much redundancy of the description is found all through the manuscript, which make it harder to follow the manuscript. Also many typo errors and grammatically incorrect expressions are found. The overall description should be tightened with the

close screening of the writing. I list just a few examples which require correction and reflect that the manuscript is not ready for the publication in its present form.

Abstract; ...revealing that graphene acts not only as mere spacer but has a direct role... → ...revealing that graphene does not act as a mere spacer but has a direct role... (*meaning completely opposite to what is intended*)

#191; Figs.3(a-e) → Figs.3(a-f)?

#198; see Fig.3(f) → see Fig.3(g)

#239; see fig supp ... → specify the figure

#264; reuniting → realizing?

Many more

Point-by-point answer to the referees reviews

We show referee questions in red italics, answers in black and added text to revised manuscript in black bold fonts, for easy identification.

Answers to Referee 1

- We highlight that Referee 1 mentioned: “[...] *this study constitutes the first graphene-based study on the similar magnetic functionality. It also provides details of the device fabrication, stability of the magnetic properties and its control, and the theoretical analysis as well, which will be much beneficial to the ones working in the related fields. Thus I can recommend accepting the manuscript for publication in Nature Communications. Nonetheless I believe the following points should be complemented before its full acceptance.*”

We thank referee 1 for her/his very positive appraisal of our results and manuscript, highlighting its novelty and significant interest for the scientific community. We appreciate her/his comments and suggestions, which we have carefully addressed point by point in our revised manuscript.

- Although authors provide surface analysis results to persuade of the high quality of their Co and Fe thin films, there is no picture showing directly their surface morphology. The STM topography, AFM or at least SEM microscopy should be provided maybe in the Supplementary Information. It is mentioned in the main text that the Co adatom intercalation took place at graphene point defects and wrinkles. This seems to suggest that much disorder was present in the graphene layer, which would have certainly been even more degraded after the Co intercalation. Thus, to validate the numerical calculation for the ideal situation more information should be provided for the surface state of the monolayer graphene right after the Co intercalation.

- Information should include the monolayer coverage of the graphene layer and the distribution of its different domains, which, according to the numerical calculation, is essential to generating the desired antiferromagnetic superexchange coupling between Co and Fe layers.

Following the referee #1 requests, we now include as supplementary material (section S3, Figure S6) a high resolution SEM study of our Graphene layers before and after the intercalation stage, together with pertinent discussion. This indicates, in line with findings by other teams and our complementary LEED study, that the intercalation process does not degrade the quality of the Graphene layer.

We thank the referee for this suggestion and consider the concerns raised by the referee as a fair and reasonable surface science questions. In the revised manuscript we cover better this point (lines 115-126) clarifying that previous related atomic-resolution microscopy investigations of the intercalated structures [refs 29,31,34], have already established and addressed the points raised by the referee. Given that structural and atomic scale aspects of the Co intercalation process have been the object of considerable experimental studies and appear well established, our work and manuscript is highly focused at exploring the perpendicular AFC across a Graphene layer and at characterizing the magnetic properties of such systems.

- As mentioned above, a previous publication [JAP 99, 08B703 (2006)] reports on closely related studies on synthetic ferromagnet and synthetic antiferromagnet structures, although graphene was not adopted for the study. This work should be cited with brief mentioning on the difference between the two studies in comparison.

We thank the referee for suggesting this reference, which we now include as ref. [8] in the manuscript, referencing to it in the introduction and concluding remarks. A discussion of some of the analogies and differences between

“classic” SAF spin valves and MTJs, and eventual graphene-based SAF structures has been added in the concluding part of the manuscript.

- The description in the manuscript in general is very loose and much redundancy of the description is found all through the manuscript, which make it harder to follow the manuscript. Also many typo errors and grammatically incorrect expressions are found. The overall description should be tightened with the close screening of the writing. I list just a few examples which require correction [...].

Abstract; ...revealing that graphene acts not only as mere spacer but has a direct role.....revealing that graphene does not act as a mere spacer but has a direct role...(meaning completely opposite [...])

We have carefully review manuscript in particular the abstract and conclusion paragraph in order to correct it for loose expressions and redundancies.

#191; Figs.3(a-e) Figs.3(a-f)? ; #198; see Fig.3(f)see Fig.3(g) #239; see fig supp ...specify the figure #264; reuniting realizing? Many more

We have corrected these “typos”. We have also reviewed the draft carefully correcting some incorrect grammar uses. We thank the referee for pointing out this and helping us to improve the quality and readability of our work

Answers to Referee 2

We thank referee 2 for her/his careful reading of the manuscript and a constructive report recognizing the soundness, quality and novelty of our results in her/his report: *“based on measurements (...) which are quite convincing”, “density functional theory (...) results in good qualitative agreement”, “this is fine original work, and an additional materials family capable of providing practical magnetic devices with perpendicular magnetic anisotropy would certainly be welcome.”*

Referee #2 also adds that *“However, in my opinion, the paper does not yet make a persuasive case for why these results are sufficiently important to justify publication in Nature Communications.[...] I suggest that the authors consider the following questions and comments.”* We especially thank the referee for providing precise criticisms together with clear questions and suggestions guiding what she/he considered necessary to clarify, address or improve on a revised manuscript.

-1. Regarding “unprecedented magnetic properties”: Many different spacer layers can provide the antiferromagnetic coupling at room temperature needed for making synthetic antiferromagnetic devices (e.g., Parkin, Phys. Rev. Lett. 67, 3598 (1991)). In what way, precisely, are the magnetic properties of the graphene devices unique? For example, if the thicknesses of the magnetic layers are scaled to make an honest comparison, how does the strength of the antiferromagnetic exchange coupling in the graphene devices (i.e., in erg/cm²) compare to what has been measured previously? The paper under review never considers this central point. If the strength falls in some new regime, what is the practical significance?

We thank the referee for pointing this out. We have now corrected our confusing statement of “unprecedented magnetic properties”, whose original purpose was to make reference to the combined properties of hybrid graphene/ferromagnetic systems with perpendicular AFC versus more “classic” systems not integrating graphene, and not claiming magnetic properties of unprecedented magnitude. We also take the referee suggestion to use energy units throughout the manuscript facilitating comparisons of the magnitudes of the magnetic properties between graphene based SAF and other SAF materials.

-The paper also tries to make “ultimately-thin” devices a selling point, but existing magnetic multilayer structures are already routinely made with layer thicknesses of 1 nm and below. Why is going down to single atomic layers advantageous or important?

We thank the referee for pointing this out. To our best knowledge very few (if any) perpendicular AFC magnetic trilayer structures, approaching three atomic layers in thickness, and with a well-defined interface have been experimentally reported so far. Therefore, Fe/Gr/Co and other related SAF structures seem interesting as a test-bench system for fundamental or applied investigations of magnetic interactions and proximity effects at the sub nanometer scale. The robust perpendicular AFC makes them even more appealing, in our opinion. Nevertheless, in the revised manuscript we have avoided to stress on the “ultimately thin” character of the structures realized and focused the discussion on the more relevant results/properties. (But if the referee suggest so, this part can be included back in the text together with these clarifying comments.)

-2. Regarding “appearing suitable for application”: This point seems dubious to me, as well. To be suitable for applications, a magnetic multilayer needs not only to have excellent magnetic properties, but it must also be capable of integration into devices with large magnetoresistance to allow read-out of the magnetic state. The only practical way of doing this at present is to make MgO-based magnetic tunnel junctions, which generally requires the use of sputtered FeCoB films with an annealing step. The use of pure Fe or Co, and the rough 3-D island growth mechanism of the Fe layer both suggest that integration into practical high-magnetoresistance tunnel-junction structures will be difficult or impossible.

-3. In the concluding paragraph, the authors make a claim about being “silicon integrable”. This is not a fair claim, because the fabrication procedure involves graphene growth at 1300 K. This is far beyond what is possible for back-end processing of Si-based devices. Even the temperature of the intercalation step (570 – 700 K) could be problematic.

We are aware of the points made here by the referee, and we agree that the claim made in the abstract of “*appearing suitable for application*” was somehow too strong. In the revised manuscript the sentence has been modified to: (line 6 of the abstract) “**[..] and gather a collection of magnetic properties well-suited for applications.**” In this way we stress that the realized system has a number of magnetic properties that match those well suited for application in perpendicular devices, but we avoid a strong statement of appearing already suitable for application which, although we think might be the case, will be more appropriate for a continuation work demonstrating a device prototype.

Regarding the second point raised and concerning the silicon integration, our original claim was related to the possibility of the monolithic growth or integration of graphene on silicon substrates, and consequently of the SAF structures described in the manuscript. We agree that the point was not properly explained and could have led to the conclusion made by the referee. To this end we modified the manuscript adding a new paragraph (lines 331-351) covering in larger detail the monolithic growth “or integration” in silicon, scalability (growth at a wafer scale) and the compatibility with fabrication processes: “**[...] Finally, it is worth briefly discussing potential paths and challenges for the eventual technological introduction of these structures on devices, reviewing aspects such as Silicon integration, scalability and compatibility with device fabrication processes. [...]**”.

We also take the opportunity to touch briefly on the interesting matter and issues for CMOS integration raised by the referee (to which we were not referring in the original draft by “silicon integration”). In doing that we note that we share the concerns raised by the referee, recalling that these are indeed rather general present challenges for graphene-hybrid technology development and graphene spintronics in general, and not merely to the case discussed here. Our viewpoint is optimistic, and we consider that there are sufficient indications -based on reported work and on-going efforts by the graphene-spintronics community, road maps, etc... - to give credit to the feasibility of graphene based technology such as spin valves, MTJs or magneto-resistive sensors and structures similar to, or even more

complex, than the ones discussed here. We have introduced this perspective and incorporated various recent and relevant references in the revised draft. We thank the referee for having raised these concerns inducing our further elaboration of these substantial aspects, which we think enriches the final part of the manuscript.

-4. The authors express the results of the DFT calculations for the magnetic anisotropy and the antiferromagnetic exchange coupling in terms of energies (2.1 – 4.5 meV for the PMA and 255 meV for the AFC). However, these quantities are really energies per unit area. What is the relevant area for interpreting these calculations?

We agree with the referee in that the exchange coupling energies given in meV are really energies per unit area. This was assumed in the definition of the J , as a difference of energy between systems with different magnetic orientations. Each of these systems has a unit cell area that corresponds to that of the graphene's overlayer. In this way, our values could be directly compared with those reported for related theoretical studies (in particular, those of Yazyez-Pasquarello [PRB 80, 035408 (2009)], or Shick and coworkers [J. Phys.: Condens Matter. 47, 476003 (2014)], but also others). We thank the referee for pointing out this issue. We have updated the units of any relevant calculated energy to mJ/m^2 . In doing so we are able to compare directly our work with other relevant theoretical, but also experimental works.

-5. The authors express the results of the experimental measurements for the magnetic anisotropy and the antiferromagnetic exchange coupling in terms of effective magnetic fields (in Tesla). They should provide the translation between these units and the energy per unit area determined in the DFT calculations so that readers can make comparisons between the two types of results.

We thank the referee for pointing this issue out. In the revised version of the manuscript we report a translation to energy per unit area of all the relevant quantities of our AFC graphene-spaced trilayer in order to enable an easier comparison with the theoretical results and with other experiments.

We found a nice agreement between theoretical and experimental measurements for the magnetic anisotropy, probably due to the fact that theoretical values are relatively insensitive to the atomic configurations taken to model the Moiré superstructure. However, the comparison for the antiferromagnetic exchange coupling is more problematic, and the theoretical prediction is substantially higher than the experimental estimate. Nonetheless, our theoretical values are compatible with previous estimations for related systems (mostly thin Co/Gr/Co or Fe/Gr/Fe sandwiches [ref.Yazyez-Pasquarello PRB 80, 035408 (2009)]). We believe that a fair comparison between theory and experiment requires considering the corrugations of the whole Moiré superlattice, as well as temperature effects that are beyond the scope of our study.

-6. A strong antiferromagnetic exchange interaction across single-layer graphene has been observed previously for relatively-thick in-plane-magnetized magnetic layers (Li et al., Phys. Rev. B, 89, 184418 (2014)). The paper under review is different in dealing with much thinner magnetic layers with perpendicular magnetic anisotropy, but they should still consider referencing this previous work.

We thank the referee for suggesting this reference, and we note that it was already included in the manuscript as ref [2]. However, we now better reference to it in the introduction and conclusion part of the manuscript as it highlights the interest of hybrid Gr/FM systems as MTJs and confirms experimentally with a device-like realization that graphene is quite effective to fabricate relatively large pinhole-free areas in magnetic heterostructure.

- 7. The paper contains several awkward uses of English. These are relatively minor in that they do not affect understanding the work, but the paper could benefit from further editing in this regard. The level of detail provided about the experimental protocols is good.

We have made careful revision of grammar and typos in our resubmitted manuscript.

Answers to Referee 3

Reviewer #3 indicated “ *The authors presented both experimental and theoretical results showing antiferromagnetic coupling (AFC) along with perpendicular magnetic anisotropy (PMA) in Fe/Gr/Co on Ir(111) structures which can serve as synthetic antiferromagnet (SAF) for spintronics. While the findings are interesting and represent a certain degree of novelty, it is hard to find them really breakthroughing in a view of previous reports in the literature including those the authors cite in the manuscript. In fact, some of the references are cited "superficially" even though they should be referred more appropriately. I think a major revision of the text, especially the introduction and conclusion sections, is needed prior to be accepted for publication in order to show/highlight the added breakthrough in the field by citing in appropriate context previous works in the topic and clearly reveal the originality of the results presented.* [...] “SAF importance in their appropriate context along with appropriate references should be given in introduction ” [...] “The importance of PMA for spintronics is missing and should be highlighted as well.”

We thank the referee #3 for highlighting some weaknesses in our manuscript. We found the comments constructive, and worthy of consideration.

We agree on emphasizing more strongly the technological relevance of SAF PMA materials, and we have modified the abstract and the introduction accordingly. We have also performed a major revision of the concluding section adding paragraphs that highlight the added breakthrough of our results and that introduce a comparison of Gr-SAF materials versus standard spin valve and TMJs structures.

We fully agree on the importance of the theoretical prediction of graphene AFC materials with PMA (in particular we refer to the report by Yang *et al.* Nano Letters 16, 145 (2016) ref 21 in the revised manuscript) and especially thank the referee for bringing to our attention the news/viewpoint article highlighting the relevance and interest of such materials, theoretically predicted. This has been emphasized in the introduction and helped to highlight the relevance of the perpendicular AFC systems here demonstrated.

With respect to the relevance of our results, and without being detrimental to the important theoretical prediction of Yang *et al.*, we claim equal relevance for the experimental realization of graphene spaced perpendicular AFC structures accompanied by supporting theoretical predictions and the significant finding of the robustness of the AFC. We hereafter highlight the principal points of interest and novelty of our work, expanding what we summarized in the letter to the editor:

-First, the relevance and value of an experimental demonstration of Gr AFC heterostructures similar to the ones recently proposed theoretically by Yang *et al.*, cannot be dismissed or taken for granted: one may just consider how in Barla *et al.*, (ACS Nano 10, 1101 (2016)), the experimental results show a clear in-plane AFM coupling for single Co adatoms (about 0.0015 ML coverage) whereas theoretical simulations predicts FM coupling for low/typical on-site Hubbard term energy $U=3$ eV. A complex scenario of FM and AFM coupling depending on absorption site was obtained only if, driven by the discrepancy with experimental results, an increased Hubbard term ($U=7$ eV) was considered. The experiments reported by Barla *et al.* yield a clear in-plane FM coupling situation at Co thicknesses above only 0.3 ML. Moreover, the corresponding theoretical calculations of Barla *et al.* predict FM coupling for a Cobalt monolayer on Gr/Ni, whereas AFM coupling for a single Ni monolayer on Gr/Ni (which was not studied experimentally). The example reported evidences that the correct theoretical prediction of AFM coupling across graphene is very delicate and can depend on various subtle small energetic balances; it suggest also that one should be careful to extrapolate results from one system to another.

-Second, Yang et al. have as well discussed the high relevance and potential technological impact of perpendicular GR-spaced AFC heterostructure, particularly as a material with high PMA and low spin orbit coupling (low damping). Our results appear novel as they provide a rather complete and original set of experiments demonstrating the robustness of AFC versus temperature and field, factors which are critical in views of potential applications. We note that the stability of the AFC versus temperature, field and Fe top layer thickness was not covered by the theoretical calculations in Yang et al, which otherwise included a very interesting calculation for the strong PMA enhancement for stacked $\text{Gr}/[\text{Co}/\text{Gr}]_m$ multilayers and the occurrence of graphene-mediated AFC between the Co layers.

-Third, our thickness-dependent and temperature-dependent studies indicate the possibility to create a proposed magnetically compensated SAF structures, which to our best understanding is an original proposition and a first experimental demonstration for a graphene based system.

-Fourth, our novel experimental results come with the corresponding theoretical simulations, adding value to the work. The simulations cover the case of Fe/Gr/Co/Ir that was not investigated before, supporting the experimental results. Moreover the simulations demonstrate, in agreement with the experiments, that a strong PMA is obtained already at the Co single-monolayer thickness in the Gr/Co/Ir system. Our simulations confirm the mechanism of Graphene mediated super-exchange coupling (earlier proposed by Yang *et al.* for perpendicular magnetization geometry), and investigate the stability of the AFC coupling versus graphene spacer thickness showing that the AFC coupling only occurs across single layer graphene.

Based on the above reasons, we are strongly convinced that our results are original and bring significant novel insights on the field, and hope for a favorable consideration for this revised manuscript by the referee.

In the following we address the specific points raised by the referee #3:

- 1) The concept of SAF, or sometimes called artificial antiferromagnets (AAF) is well-known and widely used in spintronics since 1990s when they were proposed as a reference layer for instance in spin-valve structures and not simply in the context of RKKY interlayer exchange AFC led to the GMR etc. authors refer which creates a confusion with GMR layered structures which are not at all SAF. This confusion should be eliminated and SAF importance in their appropriate context along with appropriate references should be given in introduction.

We thank the referee for this suggestion. We have reviewed the introduction and include now a paragraph covering more extensively the development of SAF structures, their use in spin-valves also adding some new pertinent references.

- 2) The importance of PMA for spintronics is missing and should be highlighted as well. Also, when discussing the results in Fig. 1 (lines 43-49), the authors should clearly cite here Ref. 28 (C. Vo-Van et al, NJP 2010) as one of the first with PMA report in Gr/Co interfaces.

We thank the referee for the note and we have modified the manuscript introduction in order to further highlight the relevance for PMA in spintronics materials. We have stressed the reference to Vo-Van *et al.*, [NJP 12, 103040 (2010)] as one of the key early works for the development of Graphene/ferromagnetic systems with PMA in the introduction, and cite it as well when we discuss PMA experimental results in figure 1 (lines 110-117).

The authors stated that they followed the intercalation approach to fabricate their structures so I suggest to cite also the work by J. Coraux et al [J. Phys. Chem. Lett. 3, 2059 (2012)] which is important in this context and comment/compare.

We thank the referee for the suggestion, we agree that the work by J. Coraux *et al* [J. Phys. Chem. Lett. 3, 2059 (2012)] should be referenced when introducing the intercalation approach. It is now included as ref [29] and referenced as a main work for the investigation of structural aspects of the intercalation approach in Gr/Ir(111) system in the final part of the introduction. It is also cited later when discussing the intercalation procedure and as one of the early works showing that the intercalation approach does not degrade the structural quality of the graphene layer.

A minor comment is Fig. 1(a),(b) and (e) should be referred in the text.

We thank the referee for her/his comment. Such reference to figures was lost on the editing of successive versions of the original manuscript, and is now included in the revised manuscript.

3) When presenting theoretical results in lines 137-138, the authors should clearly state that the AFC and PMA calculations results for Fe/Gr/Co reproduces not only their experiment, but these results reproduces and confirms earlier proposition of the same concept of Co/Gr/Co heterostructure with strong AFC and PMA by Yang et al [Nano Lett. 16, 145 (2016)] in order to boost the effective PMA. Even though authors cite this work later in the text in the different context (Ref. 36), the primary result of that work is very similar to the one proposed here and was also featured in <http://phys.org/news/2015-12-giant-magnetic-effect-benefit-spintronics.html> The authors should compare their findings with this work.

To satisfy the referee request, in our revised manuscript we have expanded reference to the results of that theoretical calculation at various points in the manuscript such as the introduction, discussion of the theoretical simulations, and concluding part. We report the related part of the introduction paragraph for clarity:

(line 28) “[...]Moreover, graphene has been reported to promote large PMA at the interface with magnetic thin-films [20,21] thus possibly serving as a building–block for perpendicular spintronic devices incorporating a spacing layer with weak spin–orbit coupling. In this context, assessing the possibility to realize exchange coupled PMA magnetic thin-films across a single graphene layer (Gr) is of primary importance towards the realization of graphene spintronic devices. This has been recently stressed by a study of H. Yang et al. [21] with a theoretical prediction of strong PMA and AFC in Gr{Con/Gr}_m multilayers.”

4) When presenting theoretical results in Fig. 2 for PMA in Gr/Co/Ir, the authors should precise if 2.1-4.5 meV range of anisotropy values are per atom or per cell and clearly confront also these values with those in earlier reports including Refs. 36 and 28 in addition to Ref. 33. In particular it would be interesting to compare the most stable stackings in different works. Also, it is very interesting to discuss whether the PMA in their structure is provided by Gr/Co or Co/Ir interface and this should be discussed also in comparison of earlier works.

We agree with this comment by the referee #3 (also remarked by referee #2). We have modified the values of the MAE to energy-per-area units [mJ/m²] that could be directly compared to previous works. Regarding the comparison with other theoretical works we have included in the manuscript a paragraph making a detailed comparison (lines 191-210). We also took this opportunity to highlight the differences with the systems discussed in previous works, and put in context the novelty and significance of our contribution. We believe that the referee’s suggestion will improve the readability of the manuscript for readers of Nature Communications

5) I found the discussion of XMCD results in Figs. 3 and 4 a little too vague including the link between temperature dependence and Fe and Co magnetization relative role, it should be rewritten more clearly.

We thank the referee for pointing this out. In the revised manuscript we reworded the discussion relative to the temperature and thickness dependence of the XMCD signal including also more details in order to clarify the paragraph.

6) The authors performed the MAE calculations with 1000-k points using SIESTA package. Since it is a recent implementation in this code, a clear demonstration of convergence as a function of k-points number is needed.

As the referee suggested, in order to check the accuracy of the fully relativistic implementation, we include in the supplementary information the tests performed for k-points sampling convergence. Of course, many previous tests and calculations were done to validate the implementation, including thorough comparisons with other methods.

All these points should be addressed in the revised version of the manuscript in order to improve the quality of the manuscript and clearly underline the originality of the results presented not by ignoring previous works but by comparing with them. It is difficult to agree without this revision that the authors demonstrated unprecedented magnetic properties as they claim in the conclusion with a superexchange coupling which they claim original even though it was already proposed theoretically for Co/Gr/Co structures in ref. 36. So the conclusion should also be revised.

We thank the referee for the remarks and suggestions.

We agree and have extensively reviewed the conclusion part to clearly highlight our results novelty and relevance.

We agree on the remark on ref [36] (Yang *et al.* Nano Letters 16, 145 (2016)) as well, and have further emphasized (we were already recognizing that, in our opinion) the value and originality of the early theoretical prediction of Yang *et al* for perpendicular AFC in Gr/[Con/Gr]_m multilayers, which is now also stressed in the introduction (see above). The increased relevance given to the work by Yang *et al.* was also helpful to clearly set the context of our findings, recalling the potential technological relevance of Gr-based synthetic antiferromagnetic/ferrimagnetic systems with large PMA and the lacking experimental realization of robust perpendicular Graphene-based AFC systems. We think that this helps to outline the relevance and originality of our results.

We disagree with the referee criticism for us “ignoring” early works. We believe we were taking special attention to recognize early works and correctly cite pioneering groups. In particular our original manuscript had an extensive list of references, with citation to almost all previous work discussing in-plane AF coupling in Graphene materials, plus many references covering leading groups working on Gr intercalated systems, including ref 36 and many other references from this group of authors. We agree, however, in that we could have made a better job at comparing our work to some of the results reported in the references. We thank the referee for pointing this out, so in the revised manuscript we now include more discussions and comparisons to previous reports.

Regarding the remark on the superexchange coupling, it was not our intention to claim originality of the superexchange coupling across Graphene as a mechanism mediating AFC. We are aware that this was discussed by various authors (which we were citing). Maybe the wording of our statement when discussing the output of the theoretical simulations was giving that erroneous impression, so we have clarified it (line 236): **“This confirms a direct role of the graphene 2D-layer in sustaining antiferromagnetic superexchange-coupling between the two magnetic films, in line with earlier propositions by Hermanns et al[22], Barla et al[23] or Yang et al[21] among others.”**

As already mentioned in our reply to referee #2, our statement for “unprecedented magnetic properties” was expressed confusingly: we wanted to remark that the experimentally realized perpendicular AFC

graphene/ferromagnetic structures could be considered to gather unprecedented properties by the fact of combining potentially appealing magnetic properties to graphene properties. We were not intending to claim that the magnetic properties on itself were of say an unprecedented magnitude. In the revised manuscript we have removed the related short statements in the abstract and conclusions paragraphs. We also added a discussion paragraph (lines 316-330) in which we present the analogies, differences and potential advantages of hybrid Gr SAF structures with perpendicular AF coupling over “conventional” SAF structures such as spin valves with Ru spacers or TMJs with MgO spacers.

There are also many typos/corrections needed: Line 41: TMJ? ; Line 209: PMI? ; Fig. 3: a), b), c), d), ... are missing and temp. dependence should be h)? Lines 251-252: SFI?

We thank the referee for pointing out these typos, which we have corrected. We have done extensive revision of the manuscript for typos and wrong English grammar uses.

Reviewers' Comments:

Reviewer #1 (Remarks to the Author):

A major revision is made in the text to respond to the criticism and to meet the comments of reviewers. One resulting drawback is that the introductory part of the paper becomes rather too long, which may make the points and the motivation of the paper defocused. On the other hand, this revision helps highlight the background and the major points of this work, and better reference the relevant works. I believe most of the points raised by the reviewers are properly responded, corrected, and added in the revised version. I recommend accepting the manuscript for publication.

Reviewer #2 (Remarks to the Author):

The paper has been improved considerably by revision. Several overblown and poorly justified claims have been removed, and the responses to Referee 3 have also resulted in a much better discussion of the previous literature. I think it is a rather close call about whether the results of the paper are sufficiently important to merit publication in Nature Communications. The mechanisms and strengths of antiferromagnetic exchange coupling and perpendicular anisotropy observed in the graphene devices are not different in any important way, as far as I can see, from previous results using other materials families. However, the paper does present a nice combination of experimental measurements and first-principles calculations. I suspect that there will be considerable interest in the paper because it does involve a new application of graphene. In the end, I lean toward recommending publication after the authors address one comment that they neglected to address properly in the first round of review.

In my comment #6 from the first round of review, I noted that Li et al. (ref. 16 in the current version of the paper) previously observed a strong antiferromagnetic exchange interaction across single-layer graphene between magnetic layers with in-plane magnetizations, and that the authors should cite this result. However, the revised manuscript only cites this previous paper in the context of its measurements of magnetoresistance, and continues to make no mention that the strong antiferromagnetic exchange interaction was also observed previously. Given that the most important new result of the paper under review is the observation of a strong antiferromagnetic exchange interaction in perpendicularly-magnetized samples, I believe it would be improper for the authors to continue to make no mention of this closely-related previous result.

The manuscript also remains in need of editing by a native English speaker, despite the fact this aspect has been improved to some degree in revision.

Reviewer #3 (Remarks to the Author):

The authors addressed most of my previous comments but there are still a few following issues to clarify prior the acceptance of the paper, mostly on reported on RKKY exchange coupling measurements and calculations:

- 1) I found Fig. 1(d) and associated conclusions problematic since the blue curve should correspond to Fe hysteresis loop while the two cartoons show inversion of Co bottom layer (in red) and not Fe one. Authors should fix/explain/clarify this.
- 2) The same blue curve in Fig. 1(d) seems to be far from saturation, i.e. corresponding to parallel alignment of magnetizations so how authors can conclude on AFM coupling strength if they have never got into required parallel configuration?
- 3) Regarding the first principle calculations of the exchange coupling, it is not clear why authors use a definition of exchange coupling as a difference of total energy of AP and P configurations. I think it should be divided by 2, i.e. $(E_P - E_{AP})/2$ instead of $E_P - E_{AP}$. This actually might even better explain their measurement result but the convention for J_{ex} should be clearly defined, if E_{ex} is J_{ex} times a scalar product then a division by 2 should be used. In fact, I would encourage the authors to introduce the exchange coupling definition and of it J_{ex} convention and answer if it is indeed 277 mJ/m² or should be twice less. Also, authors should give details of conversion of meV/atom to mJ/m² by giving their supercell lattice parameters.
- 4) Less importantly but there are still many typos which authors should fix, e.g. in line 305 etc.

After answering/fixing these issues the paper can be recommended to be published.

Answers to the referees

We show reviewers reports in red italics, answers in black and added text to revised manuscript in black bold fonts for easy identification.

Reviewer #1 (Remarks to the Author):

A major revision is made in the text to respond to the criticism and to meet the comments of reviewers. One resulting drawback is that the introductory part of the paper becomes rather too long, which may make the points and the motivation of the paper defocused. On the other hand, this revision helps highlight the background and the major points of this work, and better reference the relevant works. I believe most of the points raised by the reviewers are properly responded, corrected, and added in the revised version. I recommend accepting the manuscript for publication.

We thank reviewer #1 for his positive appraisal and publication recommendation.

Reviewer #2 (Remarks to the Author):

The paper has been improved considerably by revision. Several overblown and poorly justified claims have been removed, and the responses to Referee 3 have also resulted in a much better discussion of the previous literature. I think it is a rather close call about whether the results of the paper are sufficiently important to merit publication in Nature Communications. The mechanisms and strengths of antiferromagnetic exchange coupling and perpendicular anisotropy observed in the graphene devices are not different in any important way, as far as I can see, from previous results using other materials families. However, the paper does present a nice combination of experimental measurements and first-principles calculations. I suspect that there will be considerable interest in the paper because it does involve a new application of graphene. In the end, I lean toward recommending publication after the authors address one comment that they neglected to address properly in the first round of review.

In my comment #6 from the first round of review, I noted that Li et al. (ref. 16 in the current version of the paper) previously observed a strong antiferromagnetic exchange interaction across single-layer graphene between magnetic layers with in-plane magnetizations, and that the authors should cite this result. However, the revised manuscript only cites this previous paper in the context of its measurements of magnetoresistance, and continues to make no mention that the strong antiferromagnetic exchange interaction was also observed previously. Given that the most important new result of the paper under review is the observation of a strong antiferromagnetic exchange interaction in perpendicularly-magnetized samples, I believe it would be improper for the authors to continue to make no mention of this closely-related previous result.

The manuscript also remains in need of editing by a native English speaker, despite the fact this aspect has been improved to some degree in revision.

We thank reviewer #2 for his overall positive appraisal and publication recommendation with the request for explicitly highlighting the previous results of Li et al, ref 16. We agree it is fair to highlight more this result and now include this sentence in the introduction to do an explicit acknowledgment to Li et al AFC finding by MR:

“(line 38) In-plane AFC through a graphene layer was early observed between a Ni thin-film and Co-porphyrin molecules at temperatures below ~200K by Hermann et al.[22] as well as in small-area MTJ devices with a graphene tunnel barrier and ferromagnetic metal electrodes by Li et al.[16]”

We have carefully reviewed again the manuscript to further improve its English style, corrected it for typos and have had this proofread by a fluent English speaker colleague.

Reviewer #3 (Remarks to the Author):

The authors addressed most of my previous comments but there are still a few following issues to clarify prior the acceptance of the paper, mostly on reported on RKKY exchange coupling measurements and calculations:

1) I found Fig. 1(d) and associated conclusions problematic since the blue curve should correspond to Fe hysteresis loop while the two cartoons show inversion of Co bottom layer (in red) and not Fe one. Authors should fix/explain/clarify this.

We thank reviewer #3 for pointing out this. The cartoons in Fig.1 panel (d) were incorrectly showing inversion of Co arrows, whereas should be showing inversion of Fe arrows. We have corrected the cartoon and re-edit the figure caption to emphasize that the XMCD sign directly corresponds to the sign of the magnetization: ***“The AF-ordering of the Fe and Co layers magnetizations at remanence is directly evidenced by the sign change of the Fe XMCD spectra (f) and loop (d) between maximum and zero applied field.”***

2) The same blue curve in Fig.1(d) seems to be far from saturation, i.e. corresponding to parallel alignment of magnetizations so how authors can conclude on AFM coupling strength if they have never got into required parallel configuration?

As described in the manuscript, we consider the exchange coupling strength to be correctly experimentally estimated by the external field required to obtain a Fe magnetization collinear with the Co one having the same magnitude than at remanence. This estimation does not involve a saturated Fe magnetization but a reversed Fe magnetization of the order of 35-40% for the sample shown in Fig.1 at room temperature (whereas the Co magnetization at remanence and maximum field at room temperature is comparable, as evinced by the square-shaped magnetization loops). We accept this might be a conservative estimate for the effective exchange coupling strength. However, this definition seems an appropriate experimental estimate if one considers that the exchange coupling field is sustaining the AF configuration of remanent Co and Fe magnetizations at zero field. On the other hand, a definition based on the field required to achieve a saturated state with Fe and Co magnetizations in parallel orientation involves an effective field required to not just override the AFC interaction but any other contribution to the saturation field, thus eventually leading to an overestimation of the exchange coupling strength. Moreover, one should take into account that the exchange interaction does not have a single value but a distribution of values on a non-ideal film structure, so one can expect to experimentally semi-quantitatively determine an average effective field, which in our opinion is to some extent reflected in the remanent state at zero applied external field. All in all, our (conservative) estimate still provides at least a lower bound that is good as a quantitative estimate within around 30%-40% error and yielding notable values.

We have done a slight rewriting of the paragraph (lines 80-89) defining the effective AF field to make it clearer for the reader: ***“(line 82) Under increased negative field the Fe magnetization decreases monotonically, and eventually crosses zero aligning progressively with both the external field and the Co magnetization. In the reversed loop branch towards positive field, one finds again a remanent AF configuration with antiparallel magnetization and then attains positive coercive field, at which the Co layer reverses and drives once more the reversal of the Fe layer to maintain an AF coupled antiparallel configuration. For a larger external field that we define as the effective AFC field, $\mu_0 H_{AFC}$, the orientation of the Fe magnetization reverses and reaches a magnitude equal to its remanent value but now in parallel orientation with respect to the Co layer, thus overcoming the AF coupling interaction. This allows estimating the effective exchange--coupling energy density from the H_{AFC} determined via the Fe magnetization loop as $J_{ex}=M_{Fe} t_{Fe} \mu_0 H_{AFC}$, which might be considered as a conservative estimate because using a field smaller than the saturation field. Still, using approximated values for M_{Fe} and t_{Fe} as the Fe magnetization and layer thickness, we estimate $J_{ex}=0.74\text{mJ/m}^2$***

at $T=300\sim K$ for the Fe[1.6ML]/Gr/Co[1.9ML]/Ir(111) sample that is an exchange coupling strength of the same order of magnitude than that reported for conventional RKKY SAF multilayers [5].”

We also have slightly redesigned the sketches (cartoons) on Fig.1 to better illustrate the effective AFC field definition.

3) Regarding the first principle calculations of the exchange coupling, it is not clear why authors use a definition of exchange coupling as a difference of total energy of AP and P configurations. I think it should be divided by 2, i.e. $(E_P - E_{AP})/2$ instead of $E_P - E_{AP}$. This actually might even better explain their measurement result but the convention for J_{ex} should be clearly defined, if E_{ex} is J_{ex} times a scalar product then a division by 2 should be used. In fact, I would encourage the authors to introduce the exchange coupling definition and of it J_{ex} convention and answer if it is indeed 277 mJ/m² or should be twice less. Also, authors should give details of conversion of meV/atom to mJ/m² by giving their supercell lattice parameters.

The definition of the exchange coupling as a difference of total energy between parallel and antiparallel configurations ($J = E(P) - E(AP)$) that we defined in line 221) is one standard choice that follows the seminal work by P. Bruno [Phys. Rev. B **52**, 411 (1995)]. This approach is based on something that could be readily calculated with DFT methods, and does not rely on any assumption. An alternative take could be based in a particular model. For example, for Heisenberg’s interaction, one would obtain $E(P) - E(AP) = -2J \cdot S^2$ (at $T=0K$). However, to our knowledge there is no general agreement on the prefactor and sign relating J to the differences in energy between the two configurations (see for example Phys. Rev. B **69**, 224413 (2004), as different models will give different prefactors. Furthermore, one would also have to determine the value for S , which adds an additional degree of complexity. We believe that $J = E(P) - E(AP)$ is a direct, convenient and standard choice that allows easy comparison with other models and calculations.

On the other hand, a direct comparison between our computed J ’s and the experimentally extracted coupling is not straightforward. We recall that 277mJ/m² is not the final value to take out of the theoretical calculations. This is the largest value that we obtain for a particular atomic arrangement. In reality, the heterostructure is made of all the possible local structural arrangements given by the Moiré pattern, and for each local atomic arrangement there is slightly different coupling (J ’s reported in Figure 2b). Consequently, on average we expect a substantially reduced value for J , which takes these different J ’s into account. We believe this was clearly stated in the manuscript (lines 226-229). However, to avoid that the reader could consider this maximum value as the one resulting from the theoretical calculations, we have modified the text as follows:

“(line 226)[...] Resulting exchange coupling energies depend on the stacking and Fe adsorption site on top of graphene, with values between 15 mJ/m² and 277 mJ/m² as indicated in Fig.2a”,

The simulations were performed using the unit cell for the Ir(111) surface, placing on top the graphene layer, with intercalated Co and Fe monolayer coverage. Following the request by the referee, we now include in the methods’ section the lattice parameter used: **“ [421] The relaxed lattice parameter for the Ir(111) surface was 2.715 Å, in very good agreement with previous experimental reports. ”**

4) Less importantly but there are still many typos which authors should fix, e.g. in line 305 etc.

We have carefully reviewed again the manuscript to further improve its English style, corrected it for typos and have had this proofread by a fluent English speaker colleague.

After answering/fixing these issues the paper can be recommended to be published.

We especially thank reviewer #3 for his skillful and careful review of our draft manuscript, pointing out the error in the cartoon and providing constructive questions and suggestions. This allowed us to correct the cartoons and make the exchange coupling discussion clearer.

Reviewers' Comments:

Reviewer #2 (Remarks to the Author):

The authors have addressed my comments in a satisfactory way and I now recommend the paper for publication in Nature Communications.

Reviewer #3 (Remarks to the Author):

I can accept authors responses/corrections to my questions even though their answer about a full saturation of parallel state is not fully convincing.

I agree that the definition of exchange coupling is a matter of convention as soon as the same convention should then be used in Line 94 equation.

I still see a typos like spaces need to be inserted between words etc.

With all this said I finally recommend the paper for publication.

Answers to the referees

We thank once again the referees for their constructive suggestions along the reviewing process and for their final publication recommendation. We show reviewers reports in red italics and our answers in black for clarity.

Reviewer #2 (Remarks to the Author):

The authors have addressed my comments in a satisfactory way and I now recommend the paper for publication in Nature Communications.

We thank reviewer #2 for her/his publication recommendation and constructive suggestions that have been very helpful to improve the manuscript at various aspects (i.e., driving us to make a better comparison with existing SAF systems, discussion of how to implement these materials, etc.)

Reviewer #3 (Remarks to the Author):

1- I can accept authors responses/corrections to my questions even though their answer about a full saturation of parallel state is not fully convincing.

2-I agree that the definition of exchange coupling is a matter of convention as soon as the same convention should then be used in Line 94 equation.

3-I still see a typos like spaces need to be inserted between words etc.

With all this said I finally recommend the paper for publication.

We thank reviewer #3 for her/his publication recommendation, and for her/his detailed and constructive revision work, which has been useful to improve the manuscript.

1- We understand the point of the reviewer, and thank the reviewer for understanding this is not a central point of the manuscript nor a problem for her/his publication recommendation. As previously reported, our experimental estimation of the antiferromagnetic exchange-coupling strength does not depend on a full saturation condition, as with our element-specific hysteresis loop measurements we can estimate the external field counterbalancing the AF coupling as the field necessary to obtain a parallel configuration of Fe and Co magnetization with an Fe magnetization of equal magnitude than its value in antiparallel configuration at zero field. As stated in the manuscript, our experimentally derived values for the AFC strength are estimates with a relatively high inaccuracy (i.e. 30% or so). Hence, the non-full saturation does not affect much our discussions nor the results presented in the manuscript.

2- In line 94 equation we use the same convention as we are simply expressing that the difference between the AP and P configuration can be experimentally estimated by zeeman energy corresponding to the action of the external field on the Fe magnetization counterbalancing the exchange coupling at zero field.

3- We have verified the paper once more for further typos and corrected them.